# Topographic representation of current and future threats in the mouse nociceptive amygdala

Anna J. Bowen [1], Y. Waterlily Huang[2], Jane Y. Chen[3,4], Jordan L. Pauli[3,4], Carlos A. Campos [2] & Richard D. Palmiter [3,4]

Adaptive behaviors arise from an integration of current sensory context and internal representations of past experiences. The central amygdala (CeA) is positioned as a key integrator of cognitive and affective signals, yet it remains unknown whether individual populations simultaneously carry current- and future-state representations. We find that a primary nociceptive population within the CeA of mice, defined by CGRP-receptor (Calcrl) expression, receives topographic sensory information, with spatially defined representations of internal and external stimuli. While Calcrl+ neurons in both the rostral and caudal CeA respond to noxious stimuli, rostral neurons promote locomotor responses to externally sourced threats, while caudal CeA Calcrl+ neurons are activated by internal threats and promote passive coping behaviors and associative valence coding. During associative fear learning, rostral CeA Calcrl+ neurons stably encode noxious stimulus occurrence, while caudal CeA Calcrl+ neurons acquire predictive responses. This arrangement supports valence-aligned representations of current and future threats for the generation of adaptive behaviors.

The central amygdala (CeA) has been identified as a key integrator of multimodal stimuli to flexibly promote adaptive behaviors based on experience and current state. The CeA is a downstream partner of cortical circuits that convey learned associations (e.g., insula, basolateral amygdala)[1–4], diencephalic structures that relay real-time exteroceptive signals (e.g., thalamus)[5,6], and hindbrain populations that transmit signals from the viscera (e.g., parabrachial nucleus)[7–9], and serves as an interface to accomplish multisensory behavioral gating[10]. Within the CeA, neuronal populations have been identified that promote learned and innate responses to appetitive or aversive stimuli, yet our understanding of how these populations represent sensory modality and give rise to behavior across time is incomplete.

Early work identified individual CeA subnuclei and extrinsic connections as important contributors to distinct processes including ingestive and fear-related behaviors[11,12], while more recently,

genetically identified cell types and their intrinsic inhibitory microcircuits have emerged as functional units. Specifically, while the CeA was first described as a visceromotor output center by early anatomists due to rich connections from its medial and lateral subnuclei (CeM and CeL) to hindbrain visceromotor centers[13,14], genetically identified populations within the CeL have lately been recognized to bidirectionally control learned locomotor responses to threats via competitive inhibitory connections[15–19] and contribute to appetitive ingestive behaviors[20–22], while neurons in its capsular subnucleus (CeC) have been defined by their response to noxious somatosensory information[21,23]. These distinct population-level functions are known to arise from a combination of distinct excitatory inputs conveying sensory or learned context, local inhibition, and extrinsic connections that accomplish motor responses[10,24–27]. Crucially, components of the CeA that give rise to adaptive behaviors may vary based on spatial location,

[1]Department of Biological Structure, University of Washington, Seattle, WA 98195, USA. [2]UW Medicine Diabetes Institute, Department of Medicine, University of Washington, Seattle, WA 98109, USA. [3]Howard Hughes Medical Institute, University of Washington, Seattle, WA 98195, USA. [4]Department of Biochemistry, University of Washington, Seattle, WA 98195, USA. ✉e-mail: abowen5@uw.edu; palmiter@uw.edu

given their dependence on subnucleus organization which varies across the rostro-caudal axis of this elongated structure.

Functionally and literally (capsular, lateral, medial), the CeA has largely been classified in the medio-lateral orientation. To examine putative rostro-caudal topography in the CeA, we focused on CGRP-receptor (Calcrl)-expressing neurons that are downstream from para-brachial negative-valence neurons[28], reside in the CeC and CeL, and respond to both somatic and visceral nociceptive stimuli[21,29,30]. CeL PKCδ+ neurons implicated in fear conditioning[17,19], nociception, and chronic pain[31] in the CeA partially overlap with Calcrl+ neurons caudally[21] but are rare rostrally[29]. Calcrl+ neurons are the only described population that extends the entire length of the CeA, and due to variability in subnucleus organization across the rostro-caudal axis are primarily capsular in rostral regions and entirely lateral in the most caudal region[21,29]. By taking advantage of the elongated CeA, we discovered a preferential topographic representation of external somatosensory stimuli by Calcrl+ neurons in the rostral CeA (rCeA) and internal sensory stimuli in the caudal CeA (cCeA). This allowed us to ask how these populations utilize distinct connectivity to contribute to behavioral responses to current and potential threats.

## Results

### CeA Calcrl+ neuron extrinsic connections support topographic sensory representations

CeA Calcrl+ neurons have been defined by their access to noxious somatic and visceral stimuli via excitatory inputs from PBN CGRP+ neurons[29,30], but whether their integration into brain-wide circuits is spatially ordered remains unknown. To assess whether CeA Calcrl+ neurons receive input from upstream brain areas that could underlie functional topography, we utilized monosynaptic retrograde tracing to fluorescently label input regions followed by whole-brain atlas registration[32] (Fig. 1a, Supplementary Fig. 1a). Detailed examination of injection site, starter-cell location and proximal retrogradely labeled cells revealed that the distribution of Calcrl+ starter cells across the rostro-caudal axis varied by subject, as intended by injections being placed at either rostral or caudal poles of the CeA AP axis (centered at -1.0 AP for rostral and -1.75 for caudal; Fig. 1b).

Following whole-brain registration of the retrogradely traced neurons, we examined gross connectivity patterns (Fig. 1c, d, Supplementary Fig. 1b–d). Most neurons projecting to CeA Calcrl+ neurons were in central structures along the anterior-posterior axis, with populations targeting cCeA Calcrl+ neurons biased more caudally; otherwise, most labelled cells were located laterally from midline, ipsilateral to the injection site, in ventral structures (Fig. 1d). Notably, the most posterior input population was in the pons (PBN, Supplementary Fig. 1b), with no cells identified in either the medulla or dorsal horn of the spinal cord. Examining input distribution by macrostructure, most labelled cells were in cortical areas (cortical plate and subplate combined > 50%), with striatal structures making the next largest contribution (~25%) (Fig. 1e). We identified 40 brain structures (Allen Mouse Brain Atlas taxonomy) providing substantial input to CeA Calcrl+ neurons, including basomedial and basolateral amygdala (BLA and BMA), hippocampus (dentate gyrus, DG; subiculum, SUB; entorhinal area, ENT), olfactory cortical areas (piriform and postpiriform transition area (PIR, TR); cortical amygdala, COAp), insular cortex (visceral and gustatory areas, (VISC, GU); posterior agranular insula, AIp), temporal association cortex (TEa), caudate putamen (CP), substantia innominata (SI), and local connectivity within the CeA (Fig. 1f, Supplementary Fig. 1c, d).

To determine a spatial organization of PBN inputs, we looked for a relationship between CeA starter cells at each anterior-posterior (AP) level and resulting cell numbers in the PBN and found that PBN connectivity was most strongly predicted by injections that targeted the caudal third of the CeA, consistent with spatially biased inputs (Supplementary Fig. 1e–g). To identify similar relationships among other

input structures, we utilized hierarchical clustering and found eight primary clusters of upstream structures whose cell numbers varied together across subjects (Fig. 1g, Supplementary Fig. 1h). Subsequent examination of mean relationship between starter-cell locations and penetrance for each cluster across subjects revealed four clusters that were related to starter cells located caudally in the CeA, while a single cluster was predicted by starter cells located rostrally (Supplementary Fig. 1i). Notably, the rostral cluster contained brain regions associated with exteroceptive sensory modalities and multimodal association, including the SPFp, BLAa, and SSs (supplementary somatosensory area), while the caudal clusters contained all the viscerosensory inputs including insula (VISC, GU, AIp), PBN, BLAp, and multiple olfactory areas (PIR, TR, COAa, COAp) (Fig. 1g).

While our whole-brain registration identified differential BLAa vs BLAp inputs depending on starter-cell location, we more directly assessed the spatial organization of BLA to CeA circuitry by correlating BLA cells per section to CeA starter-cell number in the same brain section. We found a strong positive relationship when compared to AP-axis-shuffled data (Fig. 1h). These data suggest that BLA inputs to CeA Calcrl+ neurons are spatially ordered, with rostral BLA neurons projecting in-plane to rCeA Calcrl+ neurons (Fig. 1i), an organization that maintains the topographic functional specialization in the BLA. Because BLA neurons that encode noxious somatic stimuli are biased rostrally[33], this arrangement suggests that rCeA Calcrl+ neurons receive preferential somatic input from the BLA.

Given the importance of local CeA inhibitory microcircuitry in shaping activity dynamics and consequent function, we examined whether local CeA interconnectivity was predicted by starter-cell location. Regressing neighboring retrogradely labeled neurons to starter-cell-subnucleus distribution across subjects, we found that CeC Calcrl+ neurons receive substantial input only from unidentified CeL and SI neurons, while CeL Calcrl+ neurons were innervated by unidentified CeL and CeC populations (Fig. 1j). These data reveal that Calcrl+ neurons both receive substantial input from unknown CeA neurons and are contacted by distinct sensory modalities by spatial location.

We were curious whether differences in upstream input to Calcrl+ CeA neurons would be reflected by another indicator of cell type, so we used fluorescent in situ hybridization to assess expression profiles of 9 transcripts in the rostral vs caudal CeA (Supplementary Fig. 2a). As previously described[29], we found that CeA neurons expressing *Prkcd* were concentrated caudally; these neurons tended to co-express *Calcrl* (Supplementary Fig. 2b, c). We also discovered that the rostral CeA had more cells expressing *Drd2*, while the caudal had more cells expressing *Tacr1* (Supplementary Fig. 2b). Consistent with this, rCeA and cCeA Calcrl+ neurons have distinguishable genetic identities: rCeA *Calcrl*-expressing neurons tended to co-express *Drd2* and *Chrm5*, while those in the cCeA tended to co-express *Prkcd* and *Tacr1* in addition to *Chrm5* (Supplementary Fig. 2d, e). These findings support rCeA and cCeA Calcrl+ neurons as distinct populations, based both on topographic inputs and gene expression.

The presence of topographically distinct inputs to rCeA and cCeA Calcrl+ neurons motivated us to examine whether these populations are joined by intrinsic inhibitory connections which are present across many CeA populations that engage different sensorimotor systems[10]. We paired 60 nl, targeted injections of AAV1-DIO-ChR2:YFP in Calcrl+ neurons in the rostral or caudal poles of the CeA with expression of a fluorescent reporter (AAV1-DIO-mCherry) in the opposite pole, then sliced and recorded from Calcrl+ neurons in the reporter-expressing area and tested for interconnectivity by delivering trains of light (Fig. 1k, Supplementary Fig. 3a, b) that should activate terminals of any rostral- or caudal-projecting Calcrl+ neurons expressing ChR2 and cause IPSCs in connected neurons (Supplementary Fig. 3d, e). We found directionally biased connectivity: over half (13/20) of rCeA Calcrl+ neurons received inhibitory synaptic input from cCeA

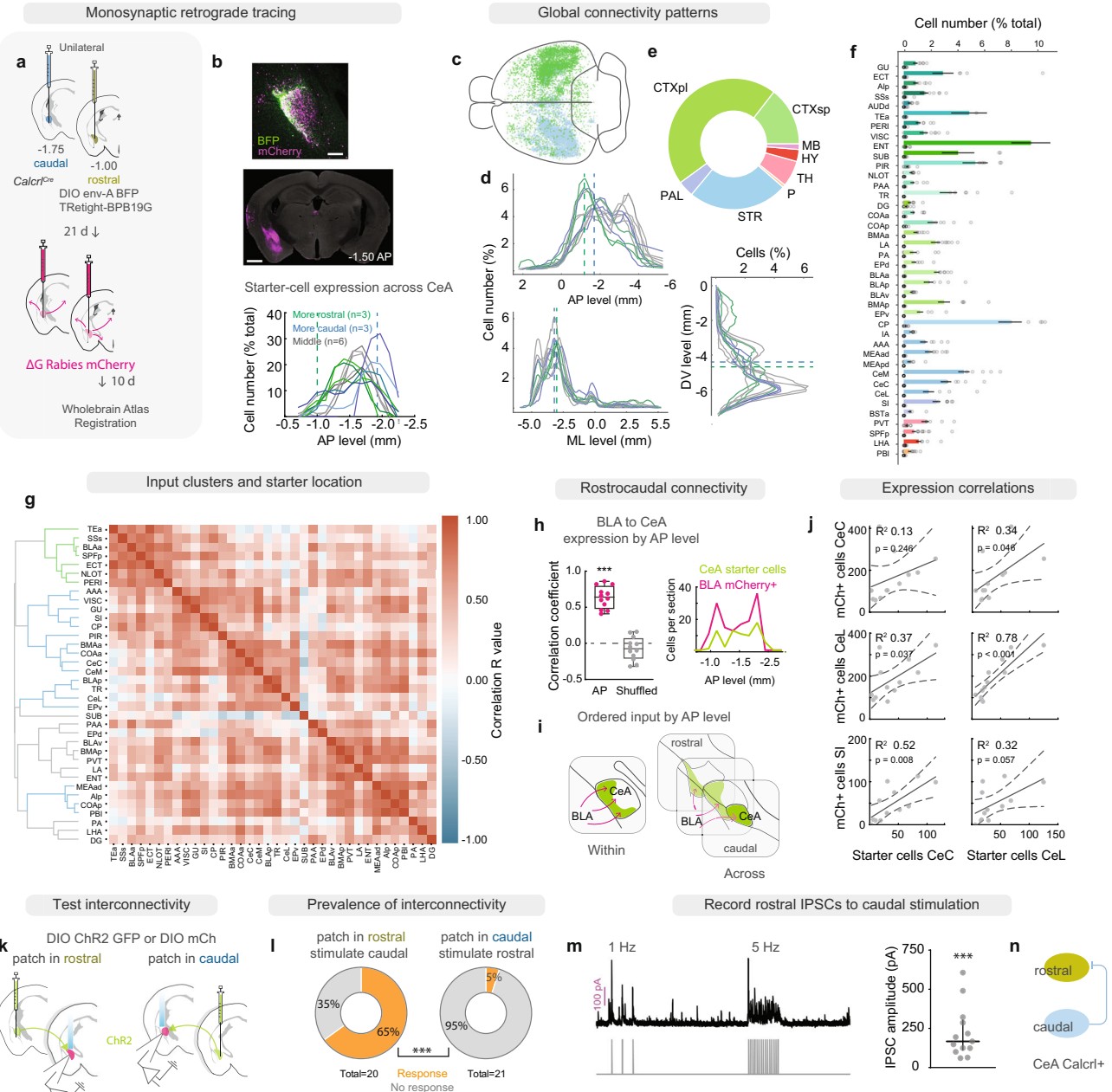

**Fig. 1 | CeA Calcrl+ neurons receive spatially segregated sensory inputs.**
**a** Monosynaptic retrograde tracing from CeA Calcrl+ neurons. Helper virus expression (green) and ΔG Rabies-mCherry (magenta) at injection site (top, scale bar 100 μm) and in adjacent regions (bottom, scale bar 1 mm). **b** Starter-cell locations (n = 12), lines indicate injection targets. **c** Whole-brain atlas registration of rabies-expressing cells. **d** Spatial distributions of traced cells. **e** Cell counts by macrostructure. **f** Regional cell counts (n = 12). Upper/lower bars: ipsilateral/contralateral counts. Data represented as mean ± SEM. **g** Hierarchical clustering of input connectivity correlation matrix with secondary correlations to starter cell location reveals rostral (green, 1 cluster) vs caudal (blue, 4 clusters) input clusters. **h** Correlation between starter-cells and BLA cell counts by AP level (two-sided Wilcoxan matched-pairs signed-rank test, p = 0.0005; centre at median, box bounds 25th and 75th percentiles, whiskers minima and maxima). **i** BLA-CeA connectivity schematic. **j** Relationships between starter-cell number, location and local connectivity (n = 12;

two-sided linear regression analysis; CeC to CeC Calcrl (top left): p = 0.246; CeC to CeL Calcrl (top right): slope = 1.9, p = 0.046; CeL to CeC Calcrl (mid. left): slope = 1.6, p = 0.037; CeL to CeL Calcrl: (mid. right) slope = 2.45, p = 0.0007; SI to CeC Calcrl (bot. left): slope = 0.83, p = 0.008; SI to CeL Calcrl (bot. right): p = 0.057). Lines and dashed bands: regression line and 95% confidence interval for slope. **k** Schematic for testing interconnectivity. **l** Proportion of rCeA Calcrl+ neurons inhibited by cCeA Calcrl+ neurons (left, 20 neurons, 4 mice); proportion of cCeA Calcrl+ neurons inhibited by rCeA Calcrl+ neurons (right, 21 neurons, 4 mice) (two-sided Mann-Whitney test, p = 0.0001). **m** Example IPSCs in rCeA Calcrl+ neuron during activation of cCeA Calcrl+ neurons (left; V_hold +10 mV); rCeA Calcrl+ neuron IPSCs during cCeA photostimulation (13 neurons, 4 mice; line at median, bars interquartile range; two-sided Wilcoxan Signed Rank test, p = 0.0002). **n** CeA Calcrl+ neuron interconnectivity. See also Supplementary Fig. 1. Full statistical information see Supplementary Table 1; area abbreviations see Supplementary Table 2.

Calcrl+ neurons, while few caudal neurons received input from rostral Calcrl+ neurons (1/21; Fig. 1l, m). These data reveal that rostral and cCeA Calcrl+ neurons not only receive distinct sensory inputs, but also are joined by directional inhibition (Fig. 1n), with caudal neurons preferentially inhibiting rCeA Calcrl+ neurons.

The spatial segregation of sensory inputs to CeA Calcrl+ neurons led us to examine whether they in turn give rise to topographically distinct outputs that could support sensory-driven behaviors. We used 100 nl, unilateral injections of AAV to express anterograde fluorescent labels (AAV1-DIO-Synaptophysin:GFP and AAV1-DIO-Synaptophysin:mCherry)

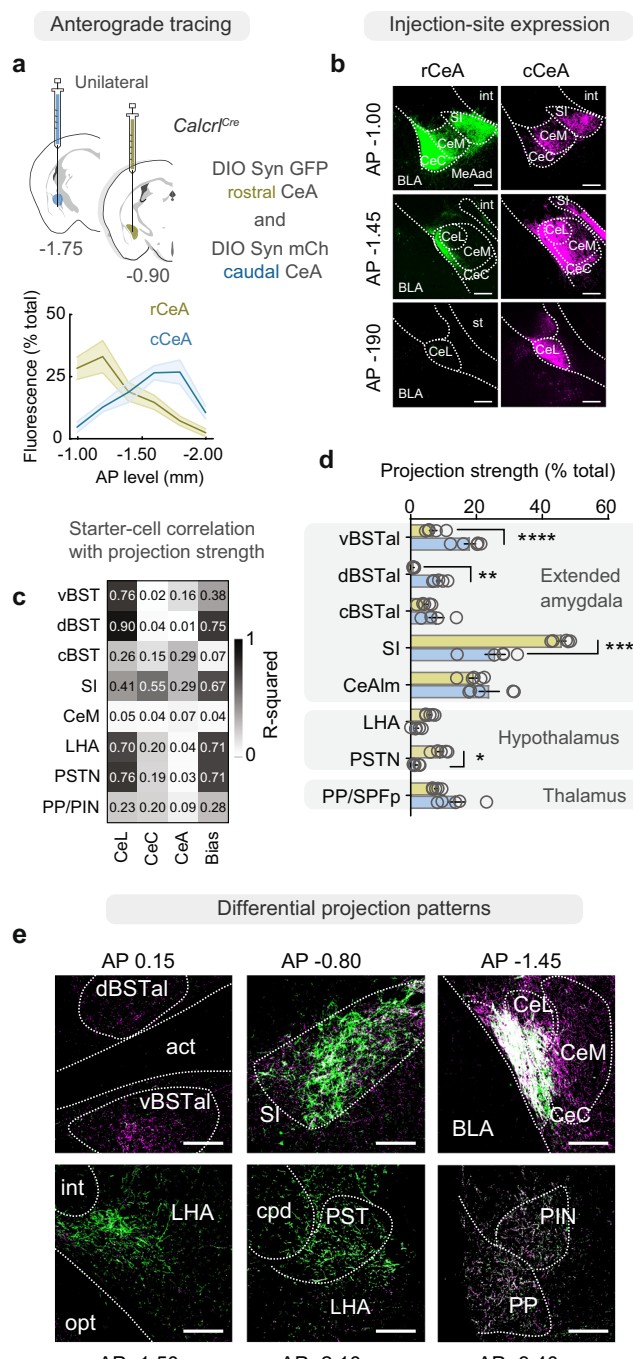

**Fig. 2 | Rostral and caudal CeA Calcrl+ neurons have different projection patterns. a** (top) Unilateral injections of anterograde tracers (AAV1-DIO-Syn:GFP and mCherry) into the rCeA or cCeA of *Calcrl^{Cre/+}* mice; (bottom) differential expression across the rostro-caudal axis of the CeA. **b** Complementary expression of anterograde tracer in CeA Calcrl+ neurons from rostral vs. caudal tracer injections ($n = 5$ ea. group). **c** Starter-cell location correlations with downstream projection strength ($R^2$ values for linear fits between variables). Bias is the slope of fluorescence expression across AP levels for each subject (e.g., −1 is from an injection with a strong rostral bias). **d** Expression of anterograde tracer in downstream targets ($n = 5$ ea. group, two-way ANOVA with Sidak's multiple comparison; $p < 0.0001$; vBSTal ****$p < 0.0001$, dBSTal **$p = 0.009$, SI ****$p < 0.0001$, PSTN *$p = 0.014$. Data represented as mean ± SEM). **e** Visualization of fluorescent tracer expressed in rostral (green) or caudal (magenta) CeA Calcrl+ neurons in ipsilateral efferent projection sites. Dashed lines indicate boundaries of white matter or nuclei. No contralateral projections were observed. int, internal capsule; st, stria terminalis; d/vBSTal, dorsal/ventral BSTal; opt, optic tract; cpd, cerebral peduncle. Scale bar: 100 μm. See also Supplementary Fig. 2. For full statistical information see Supplementary Table 1; for additional area abbreviations see Supplementary Table 2.

either promoting or suppressing activity in cCeA depending on the identity of their downstream contacts. Together, these data support the idea that rCeA and cCeA neurons mediate distinct functions through their complementary inputs and distinct outputs.

## Functional segregation of internal vs external threat-behavior responses

Our connectivity data demonstrated sensory topography across the rostro-caudal axis of the CeA, with rCeA Calcrl+ neurons preferentially receiving exteroceptive input and projecting to brain areas implicated in arousal and motivation. Hence, we predicted that rostral rCeA Calcrl + neurons would be involved in escape responses to external threats (e.g., somatic pain). We utilized 1-photon calcium imaging of virally expressed GCaMP6 to measure calcium transients in rCeA or cCeA Calcrl+ neurons during exposure to noxious heat (331 neurons, $37 \pm 12$ neurons per mouse, $n = 9$ mice, Fig. 3a, Supplementary Fig. 5a, b for lens placement) following extraction of background-subtracted, motion-corrected calcium signals[44]. Both rCeA and cCeA Calcrl+ neurons were activated by exposure to a 52 °C hotplate more than by exposure to a novel context (Fig. 3b–d, Supplementary Fig. 5d, e), with some neurons additionally encoding the time spent on the hotplate with monotonically increasing activity (Fig. 3e, f). Interestingly, cCeA Calcrl+ neurons were more likely to exhibit linearly increasing activity during heat exposure (Fig. 3g, Supplementary Fig. 5f, g, regression calculated from activity between $t = 0$ and $t = 60$ s relative to paw contact with hot plate), like upstream PBN CGRP neurons[28], while rCeA Calcrl+ neurons' activity was better fit by a nonlinear, saturating sigmoid function (Fig. 3e, Supplementary Fig. 5h, i).

To assess the contribution of either population to nocifensive behaviors, we silenced either rCeA or cCeA Calcrl+ neurons with a targeted injection of Cre-dependent tetanus-toxin light-chain (AAV1-DIO-CBA-GFP:TeTx, Fig. 3h, Supplementary Fig. 6a–c) and then exposed mice to a 57 °C hotplate (Fig. 3i). Silencing rCeA Calcrl+ neurons attenuated jumping responses to noxious heat while silencing cCeA Calcrl+ neurons had no effect (Fig. 3j, Supplementary Fig. 6e–g). Post-hoc correlations between viral expression and behavioral outcomes revealed that rostrally-biased injections targeting the CeC best predicted nocifensive behavioral suppression (Supplementary Fig. 6d). To understand the temporal role of rCeA Calcrl+ neuron activity in generating nocifensive behaviors we next artificially increased the activity of CeA Calcrl+ neurons while conducting pain sensitivity and behavioral tests using AAV1-DIO-channelrhodopsin 2 (ChR2, Fig. 3k, Supplementary Fig. 7a, b). Optogenetic activation of either rCeA or cCeA Calcrl+ neurons reduced tail-flick latency (Supplementary Fig. 7f), consistent with a pronociceptive effect[31]. We next examined nocifensive behaviors on a 52 °C hotplate (a lower

in rCeA or cCeA Calcrl+ neurons and three weeks later sectioned the entire brain to map their relative projections (Fig. 2a, b, Supplementary Fig. 4a, b). Fluorescent labeling was restricted to telencephalic and diencephalic structures, including the anterolateral bed nucleus of the stria terminalis (BSTal), dorsomedial and ventrolateral basal forebrain (SI)[34], lateral hypothalamic area (LH and parasubthalamic nucleus (PSTN)), and posterior paralaminar thalamic structures (peripeduncular (PP) and posterior intralaminar (PIN) nuclei)[35] (Fig. 2d). Notably, rostral injections led to more labeling in the SI, LH, and PSTN, while caudal injections led to greater labeling within the CeM and BSTal (Fig. 2c–e, Supplementary Fig 4c–e). Notably, the BST is a multimodal region implicated in malaise[36], anxiety[37–39], and fear[40], while regions targeted by rCeA neurons (SI and LH) influence cortical arousal[41,42] and motivation[43]. The preferential input from SI to cCeA revealed in our monosynaptic rabies tracing experiments suggests that rCeA inputs to SI could be

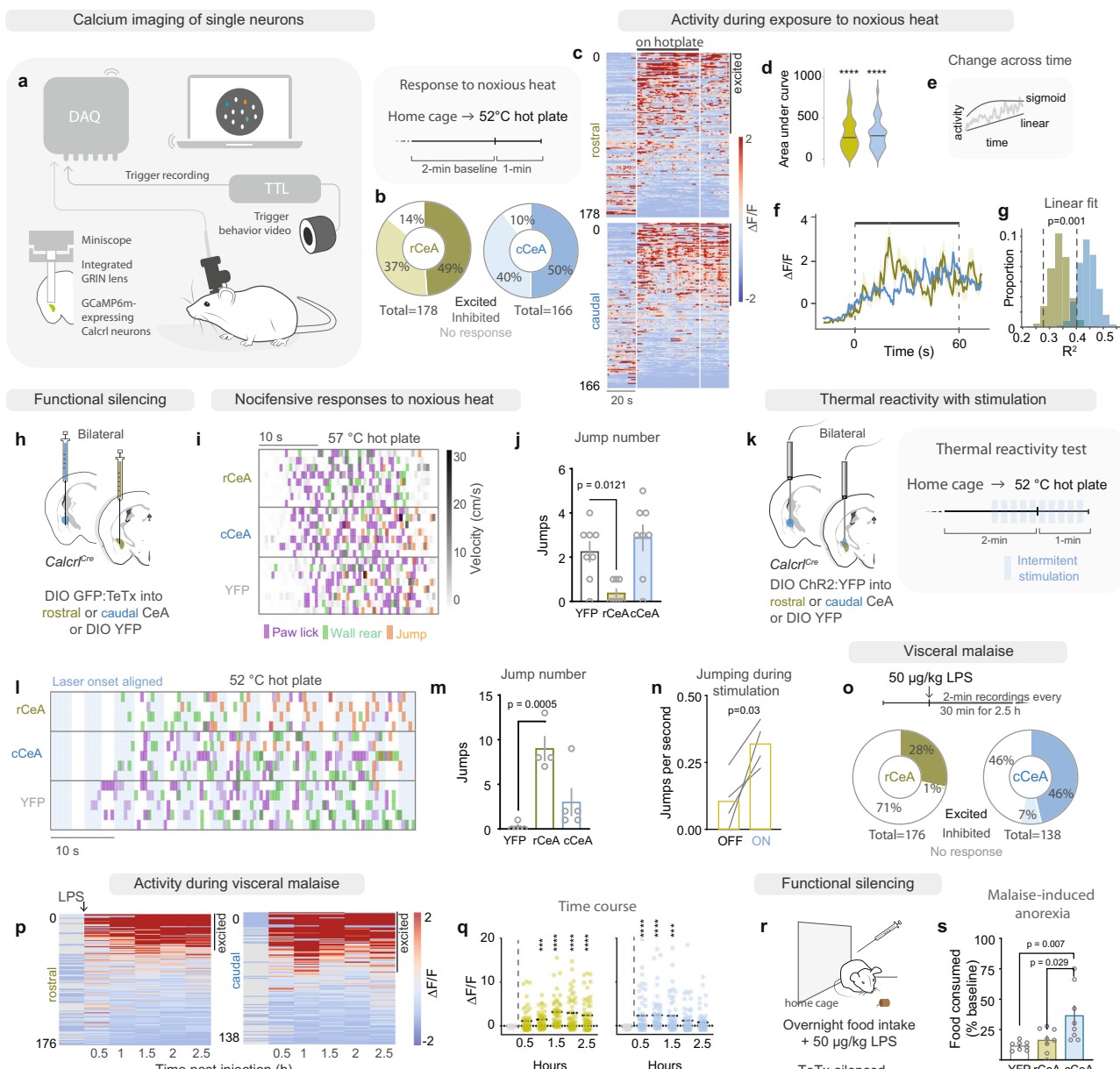

**Fig. 3 | Functional segregation of internal vs external threat behavior responses. a** Single-cell calcium imaging in freely moving mice. **b** Proportion of CeA Calcrl + neurons responding to 52 °C hot plate. **c** Activity of CeA Calcrl+ neurons on hot plate. White lines: paw contact/removal from hot plate. (n = 178 neurons rCeA, 166 neurons cCeA). **d** Activity of heat-excited neurons (area under curve; rCeA 82 neurons, cCeA 83 neurons; one sample Wilcoxan, ****p < 0.0001). **e** Linear vs non-linear fits for neural responses. **f** Activity of neurons with linearly increasing responses to hot plate (slope significantly greater than 0; n = 19 rCeA, 49 cCeA). **g** Line goodness of fit (bootstrapped distributions of R² population averages, Student's two-sided unpaired t-test, p = 0.001). **h** Functional silencing of CeA Calcrl+ neurons with tetanus toxin (TeTx). **i** Raster of behaviors exhibited on 57 °C hot plate. **j** Jumping behavior on 57 °C hot plate (n = 8 ea. group; one-way ANOVA with Dunnett's multiple comparison, p = 0.002;). **k** Photostimulation during exposure to noxious heat. **l** Raster of behaviors aligned to intermittent light delivery (blue stripes) on 52 °C hot plate. **m** Jumping response to hot-plate exposure with photostimulation (n = 5 control, 4 rostral, 5 caudal; one-way ANOVA with Dunnett's multiple comparison, p = 0.0009). **n** Jumping frequency during stimulation and non-stimulation periods (n = 4 rostral; two-sided paired t-test, p = 0.030). **o** Schematic (top) and proportion of rCeA or cCeA Calcrl+ neuron responses to LPS (bottom). **p** Neural activity in 2.5 h recording window following LPS administration. **q** ΔF/F of CeA Calcrl neurons with significant excitation (any block), across all recording blocks (lines at mean, n = 50/176 neurons rCeA and n = 64/138 neurons cCeA) (two-sided Wilcoxan Signed Rank test). **r** Paradigm for silencing rCeA or cCeA Calcrl+ neurons with TeTx and measuring LPS-induced anorexia. **s** Anorexia caused by LPS injection (n = 8 ea. group; one-way ANOVA, Tukey's multiple comparison, p = 0.006). Data represented as mean±SEM unless otherwise noted. *p < 0.05; ***p < 0.001; ****p < 0.0001. See also Supplementary Figs. 3–6.

temperature than tested with TeTx silencing that does not cause jumping in control conditions). Photoactivation caused more stimulation-locked jumping responses with expression and optic fiber placement in rCeA than in cCeA, that were absent in control mice expressing YFP (Fig. 3m, n, Supplementary Fig. 7c–e). Thus, while both rostral and caudal populations respond to noxious somatic stimuli and

affect pain sensitivity, only rCeA Calcrl+ neurons contribute to escape-like responses to external threats and aid in scaling the locomotor response to threat intensity (Supplementary Fig. 7g).

Because cCeA Calcrl+ neurons had no demonstrable role in nocifensive behaviors to a noxious external threat, we predicted that they contribute more to behavioral responses to noxious visceral

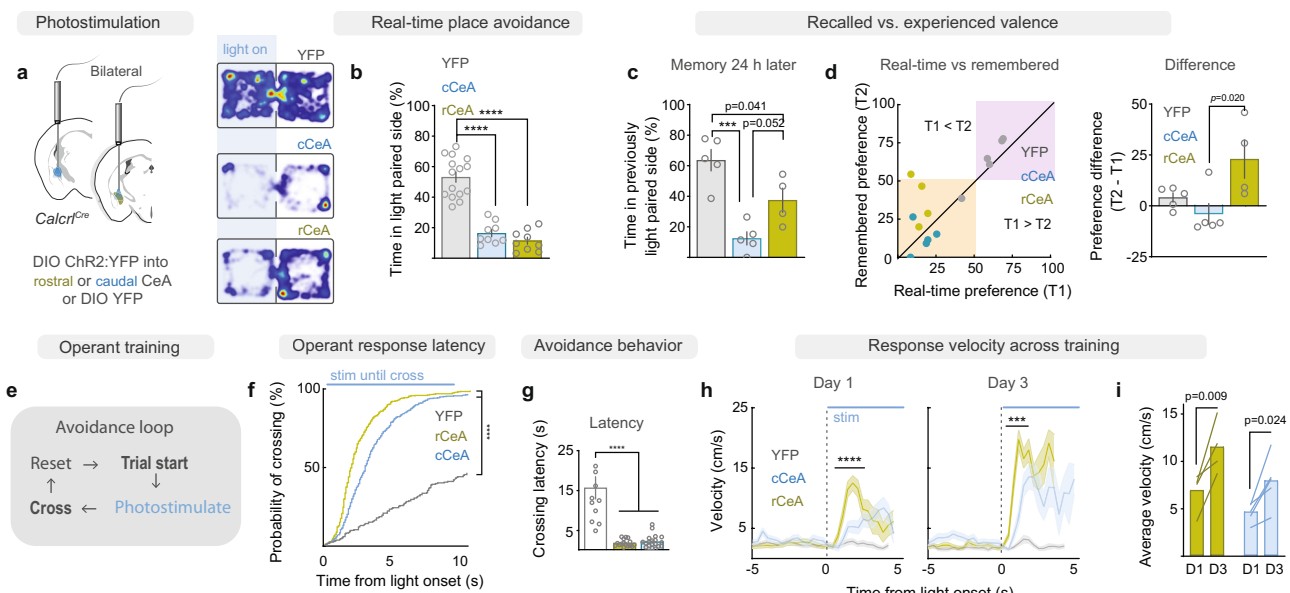

**Fig. 4 | Distinct aversive memories formed by rostral and caudal CeA Calcrl+ neurons.** a Real-time place avoidance (RTPA) paradigm. b Time spent in light paired side (*n* = 15 control, 9 rostral, 9 caudal; one-way ANOVA with Tukey's multiple comparison, *p* < 0.0001). c Avoidance memory 24 h after conditioning (*n* = 5 control, 4 rostral, 5 caudal; one-way ANOVA with Tukey's multiple comparison, *p* = 0.0004). d Relationship (left) and difference (right) between real-time and remembered avoidance (orange/purple quadrants: side-avoidance/preference) (one-way ANOVA with Tukey's multiple comparison, *p* = 0.024). e Operant avoidance paradigm: crossing sides of chamber after photostimulation initiates trial-reset with 30 s inter-trial interval. f Probability of crossing following light onset in

control vs CeA Calcrl + neuron photostimulated mice. (*n* = 5 control, 4 rostral, 5 caudal). g Latency to cross following photostimulation onset (*n* = 5 control, 4 rostral, 5 caudal; Kruskal-Wallis test with Dunn's multiple comparison, *p* < 0.0001). h Locomotion during first and third day of training ((*n* = 5 control, 4 rostral, 5 caudal; left; mixed-effects model with Tukey's multiple comparison, *p* < 0.0001. right; two-way RM ANOVA with Tukey's multiple comparison, *p* < 0.0001. i Average velocity during stimulation periods on days 1 and 3 of training (*n* = 5 control, 4 rostral, 5 caudal; two-way RM-ANOVA with Sidak's multiple comparison, *p* = 0.001). Data represented as mean ± SEM. ***p* < 0.001; ****p* < 0.0001. For full statistical information see Supplementary Table 1. See also Supplementary Fig. 7.

stimuli because they receive visceral signals and project to the BST, which has been implicated in malaise[36]. Importantly, internal threats such as inflammation induce a negative-valence state of malaise and dampen motivation normally elicited by other stimuli[45,46], suggesting these processes arise from neural circuitry distinct from acute somatic pain. However, while the generated behaviors are distinct, the populations giving rise to these outcomes have not yet been disentangled at the level of the CeA[19,20]. To induce a state of visceral malaise, we administered lipopolysaccharide (LPS; Fig. 3o), which mimics bacterial infection and causes systemic inflammation. Measuring the GCaMP fluorescence in CeA Calcrl+ neurons (314 neurons, 40 ± 13 neurons per mouse, *n* = 8 mice) at 30 min intervals following LPS injection, we found that 46% of cCeA Calcrl+ neurons were excited by LPS, peaking 1.5 h after injection, while 28% of rCeA Calcrl+ neurons were excited (Fig. 3p, q). Optogenetically activating CeA Calcrl+ neurons in fasted mice caused profound anorexia and adipsia (Supplementary Fig. 8a, b) consistent with inhibition of consummatory drive. Silencing cCeA Calcrl+ neurons with TeTx attenuated anorexia caused by LPS while silencing rCeA neurons had no effect (Fig. 3r, s). Interestingly, silencing cCeA Calcrl+ neurons affected ingestive behavior only during LPS-induced malaise, as overall homeostatic intake patterns (meal size and number, overall intake) associated with satiety signaling were unaffected (Supplementary Fig. 8c–g). Taken together, these data suggest that there is functional segregation of internal and external aversive-stimulus representations within the nociceptive CeA, consistent with its underlying spatial topography.

## Current and future-state representations in CeA Calcrl+ neurons

In addition to their selective contribution to aversive interoceptive responses, we observed that cCeA Calcrl+ neurons were robustly excited by external threats (Fig. 3d), despite not contributing to the

immediate behaviors driven by such stimuli. This suggested that their activity may contribute to some parallel function such as associative learning, given that learning involves integration of external stimuli with internal state (i.e., valence) to form future state predictions that underly learned behaviors[26,47–50]. To examine the associative signal relayed by rCeA and cCeA Calcrl+ neurons, we utilized an affective, behavior-shaping paradigm, real-time place avoidance (RTPA), where a mouse can avoid neural stimulation by moving away in a two-compartment test chamber (Fig. 4a). Interestingly, while optogenetic activation of either population of Calcrl+ neurons potently suppressed exploration of the light-paired side of the chamber consistent with promoting place avoidance and signaling negative valence (Fig. 4b, Supplementary Fig. 9a for distance moved), testing the consolidated associative memory in the absence of photostimulation the following day revealed that mice trained with cCeA Calcrl+ neuron stimulation maintained their avoidance behavior while those trained with rCeA Calcrl+ neuron stimulation had less avoidance (Fig. 4c, d). Hence, while rCeA Calcrl+ neuron activity can support immediate valuation of sensory context and generate avoidance, cCeA Calcrl+ neurons preferentially support prediction of future aversive events via association with contextual sensory information.

To more directly assess whether Calcrl+ neurons can actively promote- behaviors that minimize aversive exposure we utilized an operant-training paradigm in which a behavioral response was required to terminate photostimulation (Fig. 4e, Supplementary Fig. 9b). Mice with photostimulation of either rCeA or cCeA Calcrl+ neurons learned to switch sides to terminate photostimulation more rapidly than control mice (Fig. 4f, g), demonstrating that activity of these neurons is highly salient and can rapidly shape ongoing behavior. Both rCeA and cCeA Calcrl+ stimulation groups improved their performance across days (Fig. 4h, i, Supplementary Fig. 9c), while stimulation of rCeA Calcrl+ neurons generated more vigorous avoidance

responses both early and late in training (Fig. 4h). The combined RTPA and operant-response data indicate that while both populations generate a negative valence signal, the caudal population is better able to shape future choices through learning while the rostral population acts to promote more vigorous reactions that terminate aversive sensation.

## Rostral and caudal CeA Calcrl neurons contribute to an integrated fear-learning experience

To further examine how the dynamics of CeA Calcrl+ neurons could differentially signal current or future aversive stimuli, we analyzed their responses during auditory fear conditioning, an associative learning paradigm that requires CeA involvement[51]. During fear conditioning the sensory representation of a neutral predictive stimulus (CS, auditory tone) shifts as it comes to predict the occurrence of a noxious somatic stimulus (US, foot shock). This associative process varies in different neuronal systems, with midbrain dopamine neurons and BLA neurons individually shifting their representation from the US to the CS[52–55], while in the CeA, different populations are thought to encode the US vs CS (i.e., fear$_{OFF}$ neurons are activated by US, fear$_{ON}$ neurons are activated by CS)[16–19,56] and internally compete for action selection (i.e., running/escape vs. freezing)[16,17,56].

Having shown that rCeA Calcrl+ neurons promote escape responses to external threats, we predicted that they would contribute to US encoding during training. Measuring the activity of CeA Calcrl+ neurons during foot-shock exposure using 1-photon calcium imaging (Fig. 5a–c, analyzed 10 s before and after shock onset), we found that while rCeA and cCeA Calcrl+ neurons were equally likely to be excited by the first foot shock (Fig. 5d), rCeA Calcrl+ neurons responded more rapidly (Fig. 5e) and stably encoded the foot shock across trials (Fig. 5d, f). In support of their role in promoting escape reactions to the noxious US, we found that TeTx silencing of rCeA Calcrl+ neurons attenuated foot-shock-induced locomotion (Fig. 5g, h), while photoactivation of rCeA Calcrl+ neurons generated locomotor bursts in untrained mice (Fig. 5i, j). Together these data suggest that rCeA Calcrl+ neurons convey the current aversive stimulus and promote immediate, vigorous behavioral reactions to that stimulus. We also examined responses to the CS (10 s before and during the 20 s CS) during conditioning and found that a small proportion of both rCeA and cCeA Calcrl+ neurons encoded CS occurrence across trials, with many neurons responding to both shock and CS occurrence (Supplementary Fig. 10a–d). To identify association-encoding neurons, we focused on neurons whose CS responses changed across conditioning (Fig. 5k, l). While a similar proportion of rCeA and cCeA Calcrl+ neurons initially responded to the CS, cCeA Calcrl+ neurons that increased their CS-induced activity across conditioning had a significantly greater CS response by the final trial (Fig. 5m, n). The distinct activity profiles of rCeA and cCeA Calcrl+ neurons during associative fear learning suggest they may serve complementary roles in association formation or conditioned response generation. Indeed, permanently silencing either population with TeTx dramatically attenuated conditioned freezing behavior to the CS during and after auditory fear conditioning (Fig. 5o, p, Supplementary Fig. 10e), suggesting that both populations contribute to fear memory (Supplementary Fig. 10f) in agreement with our previous results[29]. To examine whether the activity of Calcrl+ neurons during the US contributes to acquisition of fear memory, we utilized transient inhibition from expression of the inhibitory opsin GtACR2 (AAV1-DIO-CBA-GtACR2:mCherry, Supplementary Fig. 10g, h). Interestingly, photoinhibition of either rCeA or cCeA Calcrl+ neurons during the foot shock (starting 0.5 s before and lasting 4 s after shock onset) had no effect on fear learning (Supplementary fig. 10i, j). This suggests that while rCeA Calcrl+ neuron activity following the US conveys stimulus valence and promotes behavioral responses, the activity is not necessary for downstream associative processing.

To examine fear-memory consolidation, we assessed the learned CS representation in CeA Calcrl+ neurons 48 h after conditioning in a novel context and found that significantly more cCeA Calcrl+ neurons were CS excited compared to rCeA Calcrl+ neurons (Fig. 6a–c). This is consistent with cCeA neurons maintaining CS-excitation present on the final conditioning trial (from 24% to 23%), while rCeA neurons exhibited a reduction in CS-excitation following consolidation (from 24% to 13%). Similarly, more cCeA Calcrl+ neurons were activated immediately before and during cued freezing bouts (Fig. 6d, e); these freezing-activated cCeA Calcrl+ neurons were activated during the entire CS (Fig. 6f). Because mice often freeze during the CS, this led us to ask whether freezing-excited neurons encode the CS per se or the behavioral state. To address this question, we examined the activity of CeA Calcrl+ neurons during freezing behavior that occurred outside the CS (where stimulus occurrence is different but behavioral state is similar) and at times during the CS when the animal was not engaged in freezing behavior (Fig. 6g, Supplementary Fig. 11a–d). We found that only cCeA Calcrl+ neurons activated during CS-driven freezing bouts were also active during uncued freezing bouts (Supplementary Fig. 11c), but to a lesser degree than during CS-driven freezing (Fig. 6h). These behavioral state encoding cCeA neurons were more likely than rCeA neurons to maintain their excitation during the CS when the animal was not actively engaged in freezing behavior (Fig. 6i, Supplementary Fig. 11e). Together these data are consistent with cCeA Calcrl+ neurons encoding behavioral status and CS representations. In agreement with cCeA Calcrl+ neurons supporting behaviors consistent with conditioned fear, we found that artificially activating cCeA Calcrl+ neurons induced mild freezing behavior and intermittent locomotion (Fig. 6j–l), while rCeA Calcrl+ neuron stimulation generated locomotion followed by post-stimulation freezing behavior (Fig. 6l, m). Together, these data are consistent with partially separable predictive vs noxious teaching stimulus representations in the CeA, with rCeA Calcrl+ neurons primarily encoding US occurrence, while cCeA Calcrl+ neurons are initially US responsive and maintain CS responses after consolidation. Hence, while individual cCeA Calcrl+ neurons do not shift from US to the CS as memory encoding BLA neurons do[57,58], the population activity does change, consistent with their being downstream of associative processes.

We also tested whether CeA Calcrl+ neurons support autonomic responses that are part of the coordinated response elicited by expectation of a future aversive event by transiently inhibiting them during CS delivery while measuring autonomic responses in fear-conditioned mice (Fig. 6n, o, Supplementary Fig. 11f, g). Photoinhibition of cCeA Calcrl+ neurons during the CS significantly prevented both tachycardia and hyperventilation (Fig. 6p, q), consistent with their role in supporting behavioral and physiological responses to cues predicting aversive events. While the CeA has been implicated in conditioned autonomic responses during fear retrieval[11,12,59], no individual neuron population had been identified previously; here we implicate negative-valence CeA Calcrl+ neurons in generating the autonomic responses elicited by expectation of a future aversive event.

Taken together, our results highlight the complementary roles served by CeA Calcrl+ neurons during fear learning and memory and suggest that following conditioning the stimulus representation shifts from the external stimulus encoding rostral population to the internally representative caudal population, but that negative valence signaling is maintained by its representation within the collective whole, regardless of topography.

## Discussion

What stimuli are purely external, and which are internal? This question is muddied by the reality that threatening external stimuli are perceived as such because they predict internal harm. However, the differentiation appears from the elicited motor outputs: experienced internal harm can only be managed in present and avoided in the

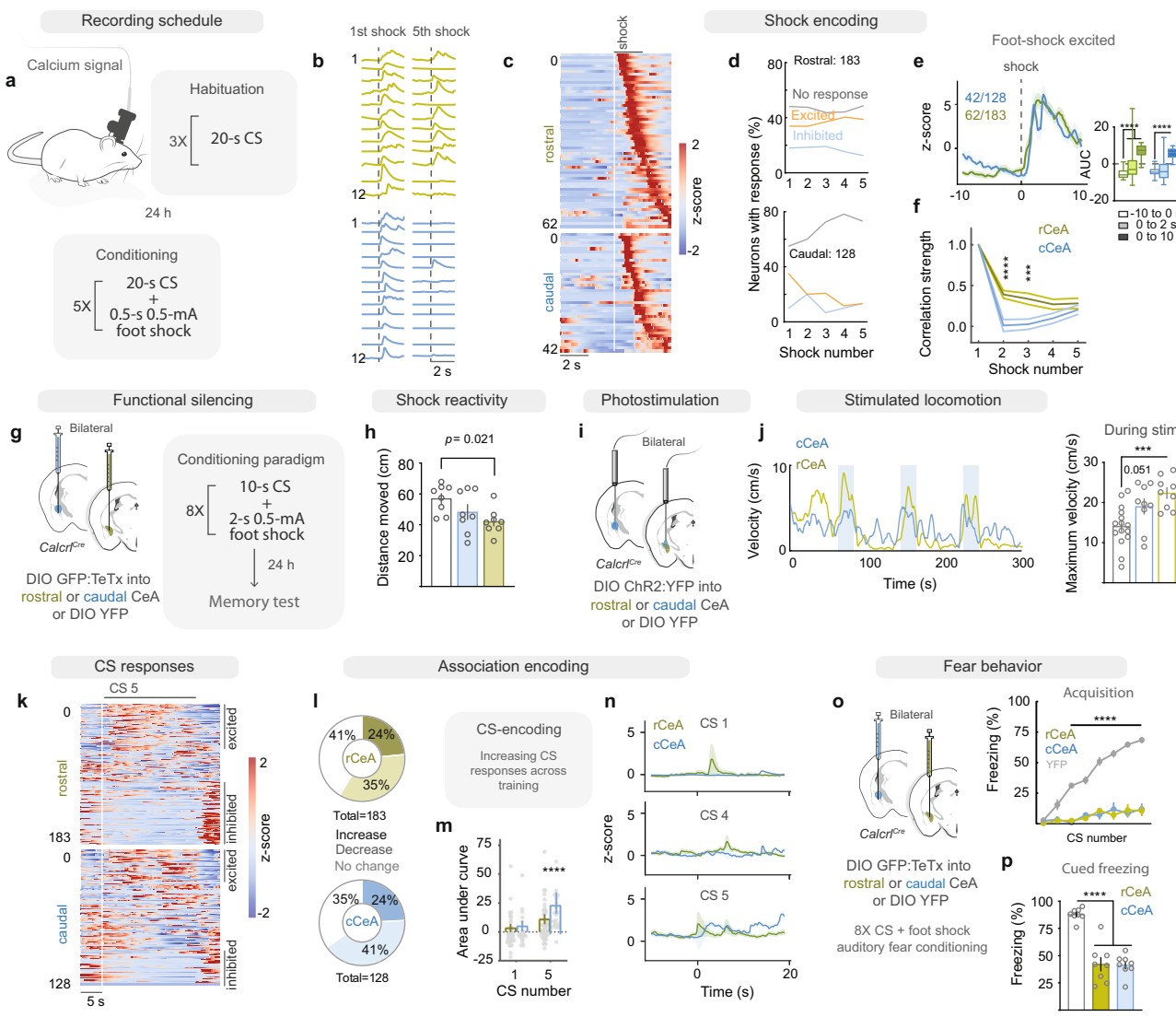

**Fig. 5 | CeA Calcrl+ neurons encode fear conditioning variables during learning.**
**a** Calcium imaging during auditory fear learning. **b** Representative calcium traces from Calcrl+ neurons during foot shock. **c** Activity of neurons excited by the first foot shock sorted by latency ($n = 128$ neurons cCeA, 183 neurons rCeA).
**d** Proportion of Calcrl+ neurons responding across 5 conditioning trials in rCeA (top) and cCeA (bottom). **e** Activity of shock-excited neurons (dotted line at foot shock onset); histograms represent baseline, shock (0–2 s), and post-shock periods (two-way RM ANOVA with Tukey's multiple comparison, $p < 0.0001$; centre at median, box bounds 25th and 75th percentiles, whiskers minima and maxima).
**f** Cross-correlated foot shock activity across conditioning in shock-responsive neurons (two-way RM ANOVA with Sidak's multiple comparison, $p = 0.0039$).
**g** Neuronal silencing prior to auditory fear conditioning. **h** Locomotor responses to foot shock ($n = 8$ ea. group; one-way ANOVA with Dunnett's multiple comparison, $p = 0.036$). **i** Photostimulation of CeA Calcrl+ neurons. **j** Smoothed group average locomotion with intermittent photostimulation (left) and average maximum

velocity during stimulation (right; $n = 15$ control, 9 rostral, 9 caudal; one-way ANOVA with Dunnett's multiple comparison, $p = 0.007$). **k** Activity of CeA Calcrl+ neurons during final CS delivery of conditioning sorted by mean response.
**l** Proportion of association encoding neurons (significant change in final vs first CS or responses that linearly change across conditioning). **m** Response to first vs final CS in all CS-excited neurons ($n = 44$ rCeA; $n = 31$ cCeA; two-way ANOVA, with Sidak's multiple comparison, $p < 0.0001$). **n** Responses of neurons with excitatory learned CS encoding across CS deliveries. **o** Freezing behavior during conditioning in TeTx and control mice ($n = 8$ ea. group; two-way RM-ANOVA with Sidak's multiple comparison; rostral $p < 0.0001$; caudal $p < 0.0001$). **p** Freezing behavior during conditioned fear recall 24 h after conditioning ($n = 8$ ea. group; one-way ANOVA with Dunnett's multiple comparison, $p < 0.0001$). Data represented as mean ± SEM. See also Supplementary Fig. 3 and 5. ***$p < 0.001$; ****$p < 0.0001$. For full statistical information see Supplementary Table 1.

future through learning, while ongoing external harm can be mitigated through action. We show that neurons of the nociceptive CeA are topographically distinguishable, with rCeA Calcrl+ neurons driving reactive locomotor responses to noxious external stimuli while cCeA Calcrl+ neurons respond to aversive viscerosensory stimuli, control learned valence, and produce inhibitory avoidance behaviors. Despite more than half of rCeA Calcrl+ neurons receiving inhibition from cCeA neurons, we find that both populations are activated with different temporal dynamics by most aversive stimuli. This suggests that this valence-aligned activity is primarily shaped by complementary

incoming sensory drive except in situations with internally biased noxious stimulation such as visceral pain or learned fear, which shifts activity towards the cCeA population to promote internally directed coping.

Our results complement a large body of work examining cell type- and projection-specific functions of CeA circuits in appetitive and defensive behaviors. The CeA has a rich community of cell populations noted for their cross-inhibitory connections promoting and suppressing locomotion during fear[15,60], conveying uncertainty and causing anxiety[25,48,54], promoting and suppressing nociception and

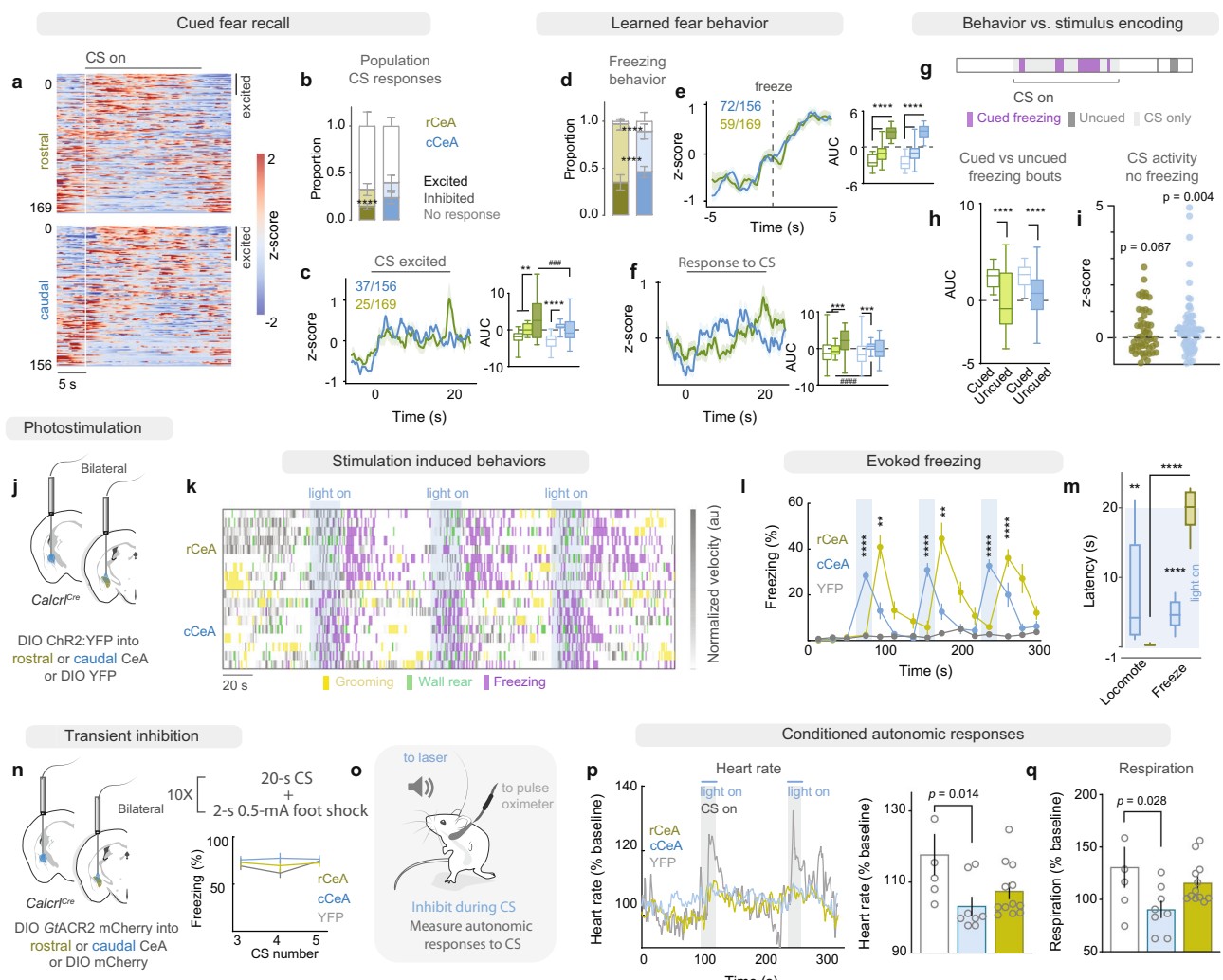

**Fig. 6 | Caudal CeA Calcrl+ neurons encode negative emotional memories.**
**a** Neuronal activity during first 3 CSs during extinction ($n = 156$ neurons caudal, $n = 169$ neurons rostral). **b** Proportion of CS-responsive Calcrl+ neurons during recall. Bootstrapped means and standard deviations for bars and error; two-way ANOVA with Sidak's multiple comparison, $p < 0.0001$. **c** Average activity of CS-excited neurons ($n = 37/156$ cCeA and $25/169$ rCeA); area-under-curve during baseline, CS, and post time periods (two-way RM ANOVA with Tukey's multiple comparison, $p = 0.005$). **d** Proportion of neurons responding during cued freezing (bootstrapped means and standard deviations for bars and error; two-way ANOVA with Sidak's multiple comparison, $p < 0.0001$. **e** Activity of neurons excited during cued freezing ($59/169$ rCeA and $n = 72/156$ cCeA; two-way RM ANOVA with Sidak's multiple comparison, $p < 0.0001$. **f** Activity of freezing-excited neurons during CS delivery (two-way RM ANOVA with Sidak's multiple comparison, $p = 0.0003$). **g** Freezing behavior during fear recall. **h** Neuron responses during cued vs. uncued freezing bouts ($59/169$ rCeA and n=$72/156$ cCeA; two-way RM ANOVA with Sidak's multiple comparison, $p < 0.0001$. **i** Z-scores of freezing-excited neurons during CS when animal is not freezing ($59/169$ rCeA and $n = 72/156$ cCeA; two-sided Wilcoxan

signed-rank test, rCeA $p = 0.0671$; cCeA $p = 0.0043$). **j** Viral injections for photostimulation. **k** Raster of behaviors during intermittent stimulation; each row is a mouse. **l** Freezing behavior during intermittent photostimulation ($n = 15$ control, 9 rostral, 9 caudal; two-way RM-ANOVA with Tukey's multiple comparison, $p < 0.0001$). **m** Latency of stimulation-evoked behaviors ($n = 15$ control, 9 rostral, 9 caudal; two-way ANOVA with Sidak's multiple comparison, $p < 0.0001$).
**n** Photoinhibition of CeA Calcrl+ neurons (left) in fear-conditioned mice (right).
**o** Measuring conditioned autonomic responses to CS-delivery. **p** Heart rate during CS delivery ($n = 6$ control, 6 rostral, 5 caudal, responses binned for each subject; Kruskall-Wallis test, $p = 0.014$; Dunn's multiple comparison). **q** Respiratory rate during CS delivery (one-way ANOVA with Dunnett's multiple comparison, $p = 0.041$). Data represented as mean ± SEM. Box plots: centre at median, box bounds 25th and 75th percentiles, whiskers minima and maxima. **$p < 0.01$; ***$p < 0.001$; ****$p < 0.0001$. Across groups comparisons requiring significant interaction: ###$p < 0.001$; ####$p < 0.0001$. See also Supplementary Fig. 6. For full statistical information see Supplementary Table 1.

conditioned freezing[18,19,31,61], and controlling ingestion and satiety[20,21,62]. A hallmark of these competing populations is their differential activation by sensory inputs which allows a sensory state to bias the competition between competitors to flexibly shape behavior[15,21]. In agreement with this pattern, we observed topographically ordered input from many brain regions depending on their spatial location, with cCeA Calcrl+ neurons preferentially receiving inputs from visceral, gustatory, and olfactory regions while rCeA Calcrl+ neurons are recipients of external modalities via thalamus and somatosensory cortex. The CGRP receptor-expressing neurons in the CeA were

previously known as one of the primary targets of the spinal-parabrachial-amygdaloid pain circuit[23,29,63–66]. Surprisingly, the PBN input is minor (< 1% of total) and cCeA Calcrl+ neurons have stronger input from the PBN[67,68] than the rCeA, in terms of cell numbers (Fig. 1f). The rCeA Calcrl+ neurons may shape circuit-level temporal dynamics depending on the phase or modality of nociception by delivering opponent inhibitory inputs to many of the downstream targets of PBN CGRP+ neurons and are themselves inhibited by cCeA Calcrl+ neurons that are functionally downstream and activated by PBN CGRP+ neurons. In this light, it is curious that rCeA Calcrl+ neurons receive

additional input from the lateral posterior thalamus, where another population of CGRP neurons reside[69]. Previous work has demonstrated the importance of CeA-midbrain connectivity for the generation of conditioned freezing behavior[24], yet despite the importance of CeA Calcrl+ neurons for this phenotype we find that they do not engage sub-diencephalic structures. Given the rich interconnectivity present across CeA subnuclei and the local projections of cCeA Calcrl+ neurons, it is possible that freezing is driven by local disinhibition of CeA neurons that project axons to the midbrain, or through alternative paths such as the LHA or BST.

Our studies relied upon spatially parsing the role of CeA Calcrl+ neurons using targeted lens insertions and viral expression, which is subject to variability and potential overlap. Our calcium imaging studies utilized a 600 μm lens, which resulted in ~200 μm of overlap in the recording plane when targeting the rostral vs. caudal extent of the CeA. Because of this, it is possible that effect sizes between groups are diminished by inclusion of the shared 'middle' population. Future work could utilize smaller diameter lenses or post-hoc registration of each field-of-view based on lens orientation to determine the true location of each recorded neuron in space. This would provide an excellent spatial map of sensory processing and reveal whether the CeA has a topographic continuum or rather, strict boundaries for internal vs. external sensory encoding.

For our optogenetic excitation and inhibition studies, we chose to use GtACR2 mediated inhibition and ChR2 mediated excitation because of both opsins are excited by blue light, which 1) has decreased penetrance relative to longer wavelengths and 2) did not interfere with pulse oximetry. Because GtACR2 causes depolarization when activated at nerve terminals[70,71], our inhibitory studies could include combined somatic inhibition and local terminal excitation. To minimize this potential, we placed optic fibers more laterally in the CeA to avoid terminals in the CeM. Given that the direction of measured effects was opposite to excitation studies, we expect that somatic inhibition was the primary effect of light delivery. For our ChR2-mediated excitation studies, we used the ChR2-H134R variant, one of the first widely disseminated ChR2 plasmids which gives rise to plateau depolarizations[72]. As such, while it reliably depolarizes the cell, it is unable to entrain depolarization to light-train frequencies, with substantial extra spikes evoked during the light train. Hence, while our studies resulted in reliable photoactivation (Supplementary Fig. 3d, e) the post-synaptic release was unlikely to exactly match our stimulation frequency, and replication studies using other ChR2 variants such as ChETA would perhaps require slightly higher frequencies than those used here.

The CeA has been proposed to mediate divergent responses by integrating external sensory information with acute internal state, i.e., its different cell populations receive sensory input biased by stimulus valence and identity and promote behaviors appropriate to the winning sensory context, based on experience and current state[10]. This model mirrors action-value signaling in striatum, where cortical state and dopaminergic input shape sensitivity to incoming sensory drive and bias cross-inhibition to promote appropriate action selection[73]. In striatal processing, a key component of action selection is a representation of sensory topography from cortex carrying stimulus location and identity. The topography of sensory inputs and function we observe across the rostro-caudal axis of the CeA suggests that appropriate action generation in the CeA also depends upon the modality of coincident sensory inputs from the cortex and elsewhere; this observation is supported by the diverging outputs from CeA Calcrl + neurons that are positioned to differentially affect arousal (via basal forebrain) and internal state (via BST), and the directional cross-inhibition that exists between them. While it could be counterproductive for a valence-assigning structure to assign opponent weights to two aversive populations, within the framework of assessing aversive-stimulus identity and appropriate action (whether coping or

locomotion) this organization is functionally beneficial[15] and helps to explain how less-salient stimuli such as visceral malaise or depression can suppress sensation and responsivity to external modalities[46,74]. A resulting open question is whether similarly ordered topographies are present in appetitive CeA populations. Appetitive ingestive behaviors transition from high-arousal, food-seeking to a lower-arousal consummatory phases and involve similar associative processes as fear learning[75]. It would be revealing if these dual processes are also topographically represented in CeA populations.

Allowing sensory topography to contribute to functional categorization aided our discovery of associations between affective processes such as valence learning and modality specialization. We were able to relate sensory processing to associative processes, and found that internally representative cCeA Calcrl+ neurons initially encoded the aversive external teaching signal prior to association formation but reduced responding as learning progressed, while shifting responsivity to the previously neutral CS. This is consistent with the neural encoding of internal state contributing to the changing value of a stimulus, rather than simply encoding its occurrence. This contrasted with rCeA Calcrl+ neurons, which persistently responded to foot shock during learning and had state-dependent effects on locomotion. Surprisingly, our data revealed that silencing either US-responsive rCeA Calcrl+ neurons or non-responsive cCeA Calcrl+ neurons attenuated fear behaviors both during conditioning and recall, despite photoactivation of rCeA Calcrl+ neurons not generating freezing behaviors. These data are consistent with the US-representation at the level of the CeA contributing to unconditioned responses, but not the fear behavior, while negative-valence, CS-responsive neurons are downstream of the associative process in BLA[68] and require consolidation for a learned representation to arise. Together these data suggest that externally biased stimuli promote arousal and reactivity, but that learning and associative processes that promote coping require a shift in neural encoding onto internally representative populations. This may serve the dual purposes of assigning a self-referential valence and promoting coping behaviors. These findings have implications for understanding phases of nociceptive processing[76] and encourage further examination of integration between sensory signaling and affective systems to provide insight into disorders of association and affect such as PTSD and depression.

## Methods

### Animals

Male and female *Calcrl*$^{Cre/+}$ mice (Han et al. 2015) were used for all studies. Following stereotaxic surgery, mice were singly housed for at least 3 week prior to and during experimentation with *ad libitum* access (unless noted otherwise) to standard chow diet (LabDiet 5053) in temperature- and humidity-controlled facilities with 12 h light/dark cycles. All animal care and experimental procedures were approved by the Institutional Animal Care and Use Committee at the University of Washington.

### Cohorts and exclusion

Photostimulation experiments consisted of two cohorts, the first of 6 cCeA (2 excluded), 5 rCeA (none excluded), and 10 control mice went through photostimulation in open field, real-time place avoidance (no recall), tail-flick latency test, and fasting-refeeding and rehydration. The second cohort consisted of 6 cCeA (1 excluded), 4 rCeA and 5 control mice that went through photostimulation in open field, real-time place avoidance with recall, hot-plate test, fasting-refeeding and rehydration, and active avoidance. These two cohorts resulted in n=9 cCeA, $n = 9$ rCeA, $n = 15$ control for overlapping studies. The tetanus toxin (TeTx) silencing experiments consisted of a single cohort of 8 cCeA, 8 rCeA and 8 control mice that went through meal-pattern monitoring, malaise-induced anorexia, hot-plate test, and auditory fear conditioning. Optogenetic inhibition (*Gt*ACR2) experiments

consisted of two cohorts, the first consisting of 8 cCeA (3 excluded), 8 rCeA (2 excluded), and 4 control mice that went through real-time place preference, auditory fear conditioning experiments, and conditioned autonomic measurements and a second cohort of 7 cCeA (2 excluded), 6 rCeA (1 excluded), and 10 controls that went through auditory fear conditioning experiments. Animals were excluded from all analyses if viral expression was weak or unilateral or if expression or optic fiber placement was outside designated boundaries. For viral expression this criterion was expression > 10% of maximum at regions posterior to −1.75 for rCeA subjects and anterior to −1.25 for cCeA subjects. For subjects with fiberoptic cannulae these criteria were expanded to include fibers centered mediolaterally over the CeA (from ± 2.90 to ± 3.25 for cCeA subjects and from ± 2.70 to ± 3.05 for rCeA subjects) and within 0.20 mm of desired AP coordinates of −0.90 mm for rCeA and −1.75 for cCeA. Animals were excluded from single experiments if they became immobilized due to patch-cord entanglement during the assay.

## Blinding
Experimenters were blinded to subject group for silencing behavioral experiments, but not during optogenetic behavioral experiments as the location of fiberoptics on the skull revealed group (slightly closer for rostral placement vs caudal) but not viral identity. To account for the possibility of placement group effects due to handling differences, rostral vs. caudal placement control groups were analyzed separately. Controls were combined for final analyses when no differences between groups were found. During video annotation analyses and cell counting/imaging experimenters were blinded to treatment group.

## Virus production
AAV1-Syn-FLEX-splitTVA-EGFP-tTA and AAV1-TRetight-mTagBFP2-B19G[77] were purchased from UNC vector core (AV6877B and AV6880B). EnvA G-Deleted Rabies-mCherry was purchased from the Salk Institute (Addgene 32635; > 5 x 10^7 viral particles/μL). AAV1-Ef1α-DIO-ChR2:YFP, AAV1-CBA-DIO-GtACR2-mCherry, AAV1-CBA-DIO-GFP:TeTox, AAV1-CBA-DIO-GCaMP6m, AAV1-hSyn-DIO-Synaptophysin (Syn):mCherry or -GFP, AAV1-Ef1α-DIO-YFP, and AAV1- Ef1α-DIO-mCherry viral vectors were produced in-house by transfecting HEK cells with each of these plasmids plus pDG1 (AAV1 coat stereotype) helper plasmid; viruses were purified by sucrose and CsCl gradient centrifugation steps, and re-suspended in 0.1 M phosphate-buffered saline (PBS) at about 10^13 viral particles/mL.

## Stereotaxic surgery
Bilateral stereotaxic injections of virus (0.06–0.28 μl per side, see below) into the rostral CeA of Calcrl^Cre/+ mice were achieved with coordinates AP -0.90 mm, ML ± 2.85 mm, DV 4.50 mm; for caudal CeA the coordinates were AP −1.75 mm, ML ± 3.10 mm, DV 4.30 mm. For functional silencing experiments 100 nl of AAV1-DIO-TeTx:GFP were injected bilaterally. For photostimulation experiments 60 nl of AAV1-DIO-CHR2-YFP were injected bilaterally. For photoinhibition experiments, 140 nl of AAV1-DIO-GtACR2-mCherry were injected bilaterally. For anterograde tracing experiments, AAV1-DIO-Syn-mCherry or -GFP were unilaterally injected into opposing poles (rostral or caudal) of the CeA (140 nl each). For monosynaptic rabies tracing experiments, we used a two-step helper virus approach that allowed us to individually titrate AAV1-Syn-FLEX-splitTVA-EGFP-tTA (1.7 x 10^11 viral particles/μL), which controls rabies virus uptake and specificity, and AAV1-TRetight-mTagBFP2-B19G (3.2 x 10^12 viral particles/μL), which allows for transsynaptic spread of the rabies virus. Pilot studies confirmed the requirement of the TVA receptor for rabies uptake and glycoprotein for transsynaptic spread. Helper viruses were mixed and 200 nl of this solution was unilaterally injected into the rCeA or cCeA and 3 weeks later 250 nl of EnvA G-Deleted Rabies-mCherry was injected. Mice were sacrificed 9 days later. For slice electrophysiology experiments, mice received bilateral injections of AAV1-DIO-ChR2-YFP (60 nl) and AAV1-

DIO-mCherry (280 nl) into opposing poles of the CeA. In mice used for optogenetic experiments, two custom-made fiber-optic cannulas were implanted bilaterally 0.2 mm above the injection site and affixed to the skull with C&B Metabond (Parkell) and dental acrylic. Mice were allowed to recover for 3 week before the start of behavioral tests.

For calcium imaging experiments, three weeks after AAV1-CBA-DIO-GCaMP6m virus injection (210 nl, unilateral) mice were anesthetized and implanted with an integrated microendoscope lens and baseplate (ProView Integrated Lens, 6.1 mm length, 0.6 mm diameter; Inscopix) that allowed for microscope attachment and visualizing fluorescent activity during the implant. The lens was targeted to -200–300 μm above the neurons using the following coordinates: rCeA AP −1.00 mm, ML −2.90 mm, and DV −4.20 to −4.50 mm; cCeA AP −1.70 mm, ML −3.20 mm, DV −3.90 to −4.20 mm. Then, a baseplate cover (Inscopix, catalogue #100-000241) was attached to prevent damage to the microendoscope lens. Post-hoc histological analysis showed variability in the lens placement relative to the rostral-caudal extent of the CeA as intended (Supplementary Fig. 4). DV was purposefully varied during lens insertions to ensure sampling from multiple DV levels of the CeA; damage due to deep insertions did not affect behavioral outcomes (Supplementary Fig. 11a).

## Photostimulation and inhibition
**ChR2**. After recovery from surgery, mice were acclimated to dummy cables attached to the implanted fiber-optic cannulas. For behavioral and autonomic studies, bilateral branching fiber-optic cables (200 μm diameter, Doric Lenses) were attached to the head of each mouse before experimentation. Light-pulse trains (10 ms) were delivered at 15 Hz as described below. Stimulation paradigms were programmed using a Master8 (AMPI) pulse stimulator that controlled a blue-light laser (473 nm; LaserGlow). The power of light exiting each side of the branching fiberoptic cable was adjusted to 10 ± 0.5 mW.

**GtACR2**. Same as above for recovery and habituation. For photoinhibition constant blue light (473 nm; LaserGlow) controlled by a Master8 (AMPI) pulse stimulator was delivered for the entirety of the inhibition window. The power of light exiting the branching fiberoptic was adjusted to 3 ± 0.5 mW.

## Slice electrophysiology
Mice were anesthetized with Euthasol (0.2 ml, i.p.) and intracardially perfused with 4–6 °C cutting solution containing (in mM): 92 N-methyl-D-glucamine, 2.5 KCl, 1.25 NaH$_2$PO$_4$, 30 NaHCO$_3$, 20 HEPES, 25 D-glucose, 2 thiourea, 5 Na-ascorbate, 3 Na-pyruvate, 0.5 CaCl$_2$, 10 MgSO$_4$. Coronal slices (250 μm) were cut with a vibratome (Leica VT1200) and kept in the same cutting solution at 33 °C for 12 min. Slices were transferred to a 25 °C recovery solution containing (in mM): 124 NaCl, 2.5 KCl, 1.25 NaH$_2$PO$_4$, 24 NaHCO$_3$, 5 HEPES, 13 D-glucose, 2 CaCl$_2$, 2 MgSO$_4$. Recordings were made in artificial cerebral spinal fluid (aCSF) containing (in mM) 126 NaCl, 2.5 KCl, 1.2 NaH$_2$PO$_4$, 26 NaHCO$_3$, 11 D-glucose, 2.4 CaCl$_2$, 1.2 MgCl$_2$ continuously perfused at 33 °C. All solutions were continuously bubbled with 95%:5% O$_2$:CO$_2$ (pH 7.3−7.4, 300−310 mOsm). Patch-clamp recordings were obtained with a MultiClamp 700B amplifier (Molecular Devices) and filtered at 2 kHz.

To assess Calcrl neuron interconnectivity across the rostro-caudal axis, post-synaptic responses to light trains were recorded from mCherry-expressing cells located in the opposite pole of Calcrl neurons expressing ChR2-YFP. Neurons were held in voltage clamp at + 10 mV and IPSCs were evoked by 10-ms pulses of blue light delivered through the objective via a 470 nm LED (ThorLabs). Events were analyzed in Clampfit v.11.0.3 (Molecular Devices).

## Histological analysis
**Tissue preparation**. Mice were anesthetized with Beuthansia (0.2 ml, i.p.; Merck) and perfused transcardially with PBS followed by 4% PFA in

PBS. Brains were post-fixed overnight in 4% PFA at 4 °C, cryoprotected in 30% sucrose, frozen in OCT compound (ThermoFisher), and stored at −80 °C. Coronal sections (30 μm) were cut on a cryostat (Leica Microsystems) and collected in cold PBS. For immunohistochemistry experiments, sections were washed three times in PBS with 0.2% Triton X-100 (PBST) for 5 min and incubated in blocking solution (3% normal donkey serum in PBST) for 1 h at room temperature. Sections were incubated overnight at 4 °C in blocking solution with primary antibodies including chicken-anti-GFP (1:10000, Abcam, ab13970) and rabbit-anti-dsRed (1:2000, Takara Bio, 632496). After 3 washes in PBS, sections were incubated for 1 h in PBS with secondary antibodies: Alexa Fluor 488 donkey anti-chicken, Cy5 donkey anti-chicken, Alexa Fluor 594 donkey anti-rabbit, and/or Cy5 donkey anti-rabbit (1:500, Jackson ImmunoResearch). The tissue was washed 3 times in PBS, mounted onto glass slides, and coverslipped with Fluoromount-G (Southern Biotech). Fluorescent images were acquired using a confocal microscope. All digital images were processed in the same way between experimental conditions to avoid artificial manipulation between different datasets.

**Anterograde tracing analysis.** Sections were collected every 180 μm across the entire brain, from caudal brainstem to olfactory bulb. To identify putative connected regions each section was visually inspected for fluorescently labeled synapses at 10X magnification. Regions considered 'connected' that were imaged and included in subsequent analysis contained more than a few brightly labeled synapses per mm$^2$ and were present across at least 360 μm in at least 2 subjects. Regions of interest were then drawn around brain regions receiving synaptic input (based on Allen Reference Atlas boundaries) and pixel intensity was measured in ImageJ and normalized to overall fluorescence in the injection site prior to cross comparison across regions and subjects.

**Monosynaptic rabies tracing analysis.** 35 μm sections were collected every 140 μm across the entire brain (from caudal brainstem to olfactory bulb) and every 350 μm from the spinal cord from T8-T10 and L2-L4. Stitched coronal images of each section were collected at 2X magnification (Keyence). Cell segmentation and forward warp registration to the Allen Mouse Brain Atlas were accomplished using Wholebrain[32] and SMART open-source software run in R 3.5.2. Not all regions with labeled cells were included in subsequent analyses as there were many with 1-2 labeled cells inconsistently present across subjects. To correct for this, we utilized an expression cutoff of 0.5%, resulting in inclusion of regions with >5 cells on average across subjects. To quantify starter cells, sections containing the CEA were incubated in blocking solution followed by overnight incubation with an Alexa647-conjugated tagBFP nanobody (1:500, Synaptic Systems, N0502-AF647). Subsequently, we acquired 10X magnification images of the CEA and we used QuPath open-source software to draw regions of interest and quantify cells co-expressing Rabies-mCherry and BFP.

**Fluorescent in-situ hybridization.** A separate cohort of C57Bl/6J mice was anesthetized with isoflurane and decapitated. Brains were extracted and frozen to approximately −30 °C on crushed Dry Ice. Brains were stored at −80 °C and then 20 μm sections were collected using a cryostat. Sections were taken every 200 μm between AP −0.6 mm and AP −2.10 mm, directly mounted onto glass slides, then stored at −80 °C. The two most rostral or caudal sections containing CeA from 3 mice were hybridized with probes for 12 mRNAs of interest in 4 groups of 3 (with intermediate fluorophore cleaving steps) using the RNAscope HiPlex assay (ACDBio); probes for *Mc4r* (melanocortin 4 receptor), *Brs3* (bombesin receptor subtype 3), *Calcrl*, *Ucn3* (urocortin-3), *Tacr3* (tachykinin receptor 3), *Calcr* (calcitonin receptor), *Prkcd*, *Slc32a1* (GABA vesicular transporter), *Chrm5* (muscarinic choline receptor 5), *Drd2* (dopamine receptor 2), *Avpr1* (arginine vasopressin

receptor 1), and *Tacr1* (tachykinin receptor 1). Fluorescent images of the CeA were taken after each hybridization step (3x3 grid at 20X magnification, Keyence BZ-X710). Images were stitched together in 4 sets of 4 channel stacks using Fiji. Probes for *Mc4r*, *Ucn3*, and *Avpr1* had weak labeling and were excluded from all subsequent analyses. Images of the probe staining within the 4-channel sets were subtracted from one another using Fiji's image calculator function to remove autofluorescence. The DAPI layers from the 4 hybridization sets were registered using the HiPlex Image Registration Software (ACDBio) and subsequently used to register all the probe images, resulting in a single 10-channel image. Individual image classifiers were trained to recognize user-identified transcript-positive nuclei for each of the 9 probes in QuPath; these classifiers were then applied to all images to identify transcript expression profiles of individual cells within the CeA.

## Behavioral measures

**Stimulation in open field.** Mice were attached to fiber-optic patch cords and allowed to habituate for 5 min in their home cage prior to placement in the arena (40 x 40 cm, white plexiglass walls). Two min after introduction to the arena, mice received 30 s photostimulation (30 Hz, 10 mW) 3 times with 60 s inter-stimulation intervals. The sessions were recorded with a USB camera attached to a personal computer and the time spent freezing (defined as immobility up until any movement of the head or body), grooming (defined as swiping movements directed towards body or face), backing, and wall rearing (both forepaws on wall) was manually scored in Ethovision (scoring was blind to treatments). Locomotor data were collected using video-tracking software (EthoVision XT 10).

**Hot-plate test.** For photoactivation experiments, mice were attached to fiber-optic patch cords and allowed to habituate for 10 min in their home cage prior to stimulation. Following habituation, mice received photostimulation (15 Hz, 3 s on 2 s off) immediately prior to placing on hot plate (< 5 s). For both activation (ChR2) and functionally silenced experiments (TeTx), mice were moved from their home cage onto the pre-heated aluminum plate (15 x 15 cm, set to 52 °C for activation experiments and 57 °C for functional silencing experiments) of a Hot/Cold Plate Analgesia Meter (Coulbourn Instruments). The transparent Plexiglas chamber (15 x 15 x 20 cm) prevented the mouse from escaping. Mice were removed after 30 s on the 57 °C hotplate or 60-s on the 52 °C hotplate. Trials were recorded with a USB camera attached to a personal computer allowing later analysis of relationship between behavior and photostimulation. Each behavioral response to the heat (paw lick, jump, and wall rear) was scored by the experimenter in Ethovision. Locomotor data were collected using video-tracking software (EthoVision XT 10).

**Tail-flick-latency test.** Mice were attached to fiber-optic patch cords and allowed to habituate for 10 min in their home cage prior to stimulation. Following habituation, mice received photostimulation (30 Hz, 8 s on/5 s off, 15 mW) for 7 min. After ending photostimulation, the mouse was restrained within a thick cloth with only its tail protruding, and its tail was partially submerged (1/2 of its length) into water maintained at 52.5 °C (± 0.2 °C). The tail-flick latency in response to heat was manually scored with a stopwatch.

**Auditory fear conditioning.** Mice underwent foot-shock fear conditioning in a shock chamber (Med Associates) over 4 days. Day 1: Mice expressing TeTx were introduced into the chamber. After free exploration of the context for 1 min, 3 CS tones (tone: 10 kHz, 20 s, 60 dB) were played at random intervals, with an average inter-trial interval (ITI) of 90 s. Day 2: Mice were allowed to explore for 1 min; then 8 CS presentations (20 s, 60 dB, 10 kHz) were played with each co-terminating with a 2-s foot shock (0.5 mA). Following the eighth CS-US pairing, mice remained in the context for 1 min before being

returned to their home cage. Day 3: Mice were returned to the conditioning context for 4 min. Day 4: Mice were placed in a novel context (same size, floor and walls replaced). After 2 min of free exploration, one tone CS was played. All the trials were recorded by a USB camera attached to the personal computer and the time spent freezing (during the CSs or Day 3 in the context), defined as immobility up until any movement of the head or body, was manually scored with a stopwatch (experimenter was blind to treatments).

**Real-time place avoidance.** The testing apparatus was a custom-made, two-chambered box (two 20 x 20 cm white poster board chambers joined by an open strip). One chamber had walls with black circles (4 cm diameter), the other was blank. Mice were attached to fiber-optic patch cords and allowed to habituate for 5–10 min in their home cage prior to introduction to the apparatus. Mice were started in the non-light paired chamber and allowed to explore freely for 2 min before the stimulation loop was started. After the habituation period, each time the mouse crossed into the stimulation chamber it received laser stimulation (15 Hz for ChR2 or 2-s on 1-s off for *Gt*ACR2) for 15 min. Behavioral data were recorded via a USB camera interfaced with EthoVision software (Noldus Information Technologies). For recall experiments mice were returned to the apparatus 24 h after the initial real-time avoidance training and allowed to freely explore for 5 min without photostimulation.

**Active-avoidance learning.** The testing apparatus was a custom-made, two-chambered box (two 20 x 20 cm white poster board chambers joined by an open strip). One chamber had walls with black circles (4 cm diameter), the other was blank. Mice were attached to fiber-optic patch cords and allowed to habituate for 5–10 min in their home cage prior to introduction to the apparatus. Mice were started in the non-light paired chamber and allowed to explore freely for 2 min before the stimulation loop was started. After the habituation period, a warning tone (10 kHz, 80 dB) was delivered. If no side-crossing occurred during the 5-s tone-only interval, photostimulation was started (15 Hz, 10 mW) and lasted until the mouse exited the chamber to the other side. If a mouse crossed sides either during the tone or stimulation, that triggered a trial reset with a 30-s ITI period during which the mouse could freely explore. At the end of the ITI the avoidance loop was reinitiated, and the warning tone was delivered. Mice received 35 min of training each day for 3 days. Sessions were controlled and behavioral data recorded by a USB camera interfaced with EthoVision software (Noldus Information Technologies). Latency to cross following warning tone delivery and velocity during tone and stimulation (if occurred) were measured.

**Ingestive behaviors.** *Meal-pattern analysis.* Mice were habituated to food-monitoring home cages (BioDAQ, v 2.2) for at least 10 days before experimental manipulation. Feeding records were analyzed using BioDAQ Viewer (v. 2.2.01). A feeding bout was defined as a meal if ≥ 0.08 g of food was ingested and if it was separated by another meal by ≥ 5 min. Mice had unlimited access to water during food-monitoring experiments. *Malaise-induced anorexia.* Mice were housed in food-monitoring home cages as described above and habituated to i.p. injections for 3 days prior to the experiment. One hour before lights out, the mice were weighed and injected with lipopolysaccharide (LPS, 50 μg/kg; Calbiochem, catalogue #437650); i.p.), and overnight food intake was measured. *Ingestion with optogenetic activation.* Mice were individually housed in clean cages without wire racks, equipped with ports on the front of the cage allowing for the insertion of a custom drinking bottle consisting of a glass test tube fitted with a drinking spout; mice were acclimated to these conditions for at least 4 days. For fast-refeed food intake measurements bedding material was exchanged and food pellets were removed 1–2 h before lights out the day before the experiment. The following morning, mice were attached to

fiber-optic patch cords and allowed to habituate for 5–10 min. Then, a single standard chow pellet was weighed and added to the cage and light was delivered (15 Hz, 3 s on 2 s off); the pellet was weighed after 1.5 h. For dehydration/rehydration experiments the water bottle was removed from the cage 1–2 h before lights out the day before the experiment. The following morning proceeded as above, except water was weighed and delivered instead of a food pellet and access lasted for 20 min.

## Calcium-imaging studies
All calcium imaging was recorded at 6 frames per second, and 0.6–1.0 mW LED power using a miniature microscope from Inscopix (nVista3.0). The recording parameters were based on pilot studies that demonstrated minimal photo-bleaching using these settings. Ethovision (Noldus, XT 10) was used to trigger and synchronize behavioral video recordings with calcium recordings. Mice were briefly (< 1 min) anesthetized with 1.5–2% isoflurane for microscope attachment for each recording session and were allowed to recover for 45 to 60 min prior experiment start.

**Order of experiments.** Hot plate > auditory fear conditioning > LPS.

**Cutaneous pain.** The hot-plate test involved removing the mouse from its home cage and placing it onto a pre-heated aluminum plate set to 52 °C for 60 s following a 2 min baseline recording taken while the mouse was still in its home cage. The recording was terminated 60 s after returning the mouse to its home cage from the hot plate. Mice underwent a 0.5 mA foot shock (0.5 s duration) that was synchronized to the calcium imaging recordings by triggering both the shock and calcium imaging acquisition software (Inscopix, nVista HD) using a TTL generated by Ethovision (Noldus, XT 10) during the auditory fear conditioning experiment described below.

**Auditory fear conditioning.** Mice underwent foot-shock fear conditioning in a shock chamber (Colbourne Instruments, 25 cm x 25 cm) over 4 days; each session started with a 1 min baseline recording in the home cage. For all recording sessions calcium imaging acquisition, behavioral videos, and delivered stimuli (cues and shocks) were all triggered by TTLs generated by Ethovision (Noldus, XT 10) to allow synchronization. Day 1: For the habituation session mice were transferred to the shock chamber and received 3 cue deliveries (10 kHz, 75 dB, 20 s duration, 90 s ITI) following a 1 min baseline recording. Separate recordings were made for each cue starting 20 s before and lasting 20 s after each cue to prevent photobleaching. Day 2: For conditioning, mice were transferred to the shock chamber and after 60 s received 5 cue pairings (10 kHz, 75 dB, 20 s duration) that co-terminated with a foot shock (0.5 mA, 0.5 s, 90 s inter-trial interval). The first cue/shock recording included the initial 60 s in the context along with the 20-s cue and 20-s post-shock period (100 s total). Cue/shock recordings 2 through 5 started 20 s before and lasted 20 s after each cue, for 60 s total each. Day 3: During the context test, mice were transferred to the shock chamber and calcium activity was recorded for 4 min. Day 4: During the cued recall session mice were transferred to a novel context (same size, walls and floors replaced) and 2 tone cues in the absence of foot shock were delivered using ITI of 60 s. Separate recordings were made for each cue starting 5 s before and lasting 10 s after each cue. Freezing behavior was quantified during the context test and during the first 2 recordings (CS 1 and 3) of the extinction test.

**Visceral malaise.** After a 2 min baseline recording, non-food-deprived mice received an i.p. injection of LPS (50 μg/kg; Calbiochem, catalogue #437650). GCaMP activity was recorded for 2 min at 0.5, 1, 1.5, 2, and 2.5 h after injection.

## Calcium-imaging processing

Calcium recording files were cropped and down-sampled (spatial binning factor of 2, Inscopix Data Processing Software) to reduce processing time and file size, and files from each recording session for each subject were concatenated chronologically using ImageJ. Then, single-cell activity was extracted using the miniscope 1-photon-based calcium-imaging signal extraction pipeline (MIN1PIPE)[44] run on MATLAB R2019a which, briefly, consists of a neural enhancing step (anisotropic diffusion, morphological opening and background subtraction) followed by a hierarchical motion correction module (using a Kanade-Lucas-Tomasi tracker for displacement estimation and diffeomorphic Log-Demons image registration for large deformations) and ending with a seeds-cleansed neural signal extraction module, which utilizes a Gaussian mixture model and recurrent neural network trained offline using a long-short term memory module to classify calcium spikes. For across time comparisons raw traces were converted to ΔF/F ($F - F_{mean}/F_{mean}$), where F was the fluorescence at any given time-point and $F_{mean}$ was the average fluorescence across the entire recording. For peri-event comparisons (e.g., activity around a freezing bout) raw traces cut around the event were converted to z-scores ($F - F_{mean}/F_{sd}$), where F was the fluorescence at each time point, $F_{mean}$ was the average fluorescence across the baseline period and pre-stimulus period, and $F_{sd}$ was the standard deviation of the fluorescence across the same window.

## Designation of responses

Calcium-imaging data were analyzed using custom code in Python. Briefly, stimuli, events, and behaviors relevant to each recording session were noted for each subject relative to recording time and were then used to select, slice, and (for freezing bouts) average activity of individual cells during peristimulus windows (10 s before/after CS and US, 5 s before/after freezing). Then, the activity (z-score or ΔF/F) of each cell relative to a pre-event baseline (for foot shocks, CS, freezing) or designated baseline period (for hot plate, malaise) were used to assess whether the cell was activated or inhibited, based on nonparametric statistical tests (Wilcoxan signed-rank test) or normalized area under the curve (> |2x| baseline period; stimulus normalized to length of baseline period). Linear and nonlinear fits for hot plate-evoked activity were calculated on ΔF/F taken from t=0 to t=60 s from first paw contact with hot plate for each neuron. Linear regression was conducted using the python scipy.stats library linregress function. For nonlinear fits we used the scipy.optimize library curve_fit function to solve for the parameters (c, k, x0, y0) minimizing error between real activity and a sigmoid function ($y(x) = c / (1 + \exp(-k*(x-x0))) + y0$). Bootstrap distributions were from 10000x random resampling of the best fit slope, R², sigmoid $k$, and change in standard error when switching from linear to nonlinear fit. Time-varying responses to CS were determined by linear regression on activity vs time with significant non-zero slope and by directly comparing first to last CS-evoked activity (Wilcoxan signed-rank test). Mean activity and normalized area under the curve for individual neurons during indicated time periods were exported and analyzed statistically in Prism 8.0 using two-way repeated measures ANOVA and multiple comparisons tests.

## Autonomic measurements

**Pulse-oximeter measurements of conditioned autonomic responses.** *Gt*ACR2-expressing mice were conditioned to asymptotic freezing with a 20 s 10 kHz CS and 2-s 0.5 mA foot shock US, as noted above (8 pairings, without inhibition). Then mice were habituated to pulse-oximetry collar sensors: mice were first habituated to dummy collar sensors (Starr Life Sciences) for 12 h overnight prior to secondary habituation to collar sensors and attached cables (Starr Life Sciences). After a full day of habituation, hair was removed from the sensor areas (circumference of neck) to allow trans-dermal infrared penetration,

and mice were switched to dummy collar sensors overnight. The next morning, collar sensors and attached cables were placed on the mice and allowed to habituate for at least 30 min prior to patch-cord attachment. Mice were then attached to fiber-optic patch cords and placed in a novel context and allowed to habituate for 1–2 h, until heart rate and respiration became stable. Autonomic measurements were taken during a 2 min baseline period and while two CSs were delivered with concurrent photoinhibition (nonstop 437 nm light, 5 mW), with a 2 min ITI. Recordings were exported and analyzed in Excel.

## Quantification and statistical analysis

All statistics and statistical tests are indicated where used. Behavioral and histological data were analyzed using Prism 8.0 (GraphPad Software), calcium-imaging data were analyzed using Python scipy.stats packages. To determine whether parametric tests could be used, the D'Agostino-Pearson Test or Shapiro-Wilk Test was performed on all data as a test for normality. Parametric tests were used whenever possible to test for differences between two or more means. Analysis of variance (ANOVA) was used to check for main effects and interactions in experiments with repeated measures and more than one factor. When main effects or interactions were significant, we conducted secondary planned comparisons according to experimental design (e.g., comparing time points). All comparisons were two-tailed. Hypothesis testing was conducted at a significance level of 0.05 and corrected where repeated measures were conducted.

## Reporting summary

Further information on research design is available in the Nature Portfolio Reporting Summary linked to this article.

## Data availability

Data generated in this study are provided in the Source Data file. Source data are provided with this paper.

## Code availability

The custom code used for analyzing calcium imaging data supporting the findings of this study are available from the corresponding author upon request.

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

## Acknowledgements

We thank L. Zweifel for insightful feedback, G. Stuber for use of electrophysiology equipment, E. Bowen for help with histology, S. Phelps for animal husbandry support, S. Ng-Evans for technical assistance and members of the Palmiter lab for helpful discussions. This work was supported in part by grants from the National Institutes of Health (NIH) (T32NS099578, A.J.B.; R01-DA24908, R.D.P.). This article is subject to HHMI's Open Access to Publication policy. HHMI lab heads have previously granted a nonexclusive CC BY 4.0 license to the public and a sublicense to HHMI in their research articles. Pursuant to those licenses, the author-accepted manuscript can be made freely available under a CC BY 4.0 license immediately upon publication.

## Author contributions

A.J.B. designed the experiments, conducted behavioral and in vivo imaging experiments, and performed all analysis. J.L.P, Y.W.H., and C.A.C. conducted histological experiments. J.Y.C. conducted and analyzed slice electrophysiology experiments. R.D.P. provided resources. A.J.B wrote the manuscript with input from C.A.C with final approval and editing by R.D.P and other authors.

## Competing interests

The authors declare no competing interests.
