## [Peer Review File · Nature Communications]

Topographic representation of current and future threats in the mouse nociceptive amygdalaEditorial Note: This manuscript has been previously reviewed at another journal that is not operating a transparent peer review scheme. This document only contains reviewer comments and rebuttal letters for versions considered at *Nature Communications*.

REVIEWER COMMENTS

Reviewer #1 (Remarks to the Author):

The authors have addressed my previous concerns and I support publication.

Reviewer #3 (Remarks to the Author):

1, I greatly appreciate that the authors plotted the locations of fiber placement for subjects. I also understand that the authors described the accuracy and efficiency of targeting the rostral vs. caudal Calcr1 population in detail, as was also in response to Reviewer #2 point #2. As they used only the extreme ends of the CeA, and excluded the animals with fibers located >200 um from the desired coordinates (page 7, 2nd paragraph), I recommend analyzing the correlation between the location of fibers and the behaviors of these animals. This will reveal if the rCeA and cCeA are two distinct sub-nuclei or a continuous structure with a subtle gradient.

2, My concern is still the definitions of rCeA and cCeA in this study. While I appreciate that the authors precisely controlled the stereotaxic coordinates, viral injection volumes, and expression levels, it still seems complicated to distinguish such small subregions. Because previous studies, including the authors' own, have demonstrated many cell-type specific marker molecules, such as PKC delta, Somatostatin, and CRF, I recommend verifying the cell types in different coordinates using immunostaining or other methods.

3, The response to Reviewer #1 point #14 was rather confusing to this reviewer. They found that the photoinhibition of either rCeA or cCeA Calcr1 neurons during the foot shock delivery had no effect on fear learning (Extended Data Fig. 9i-j). The authors argue that these results suggest that while rCeA Calcr1 neuron activity following the US conveys stimulus valence and promotes behavioral responses, the activity is not necessary for downstream associative processing. However, their previous work (Han et al., 2015) demonstrated that silencing Calcr1 neurons in the CeA significantly attenuated fear learning. Are these potentially opposing results due to methodological differences (TeTx vs. GtACR2) or potential compensatory roles of rCeA and cCeA? Does silencing both rCeA and cCeA during the foot shock attenuate the fear learning?

4, The response to my point #8 was not clear. The authors' response to my comment says, "the bulk of the cells are located quite ventrally, over 1 mm deeper than our fiber tips in this stimulation experiment. Because of this, and the variability in fiber placement that results in locations slightly rostral or caudal from this AP level, we do not think that this is the reason for the difference in behavior in the place-aversion experiment. Additionally, these same animals were used in the conditioned place aversion study where cCeA stimulation led to enhanced place aversion."

The author's group previously demonstrated that Calcr1 neurons are more in rCeA, and stimulating Calcr1 neurons in the CeA induces freezing behaviors and fear learning (Han et al., 2015). Do the authors suggest that the Calcr1 neurons stimulated in the present study are different from those in the previous work by Han et al. because they reside deeper (over 1 mm deeper than our fiber tips)? Or are there distinct sub-regions of CeA along the DV axis as well?

1, I greatly appreciate that the authors plotted the locations of fiber placement for subjects. I also understand that the authors described the accuracy and efficiency of targeting the rostral vs. caudal *Calcr1* population in detail, as was also in response to Reviewer #2 point #2. As they used only the extreme ends of the CeA, and excluded the animals with fibers located >200 um from the desired coordinates (page 7, 2nd paragraph), I recommend analyzing the correlation between the location of fibers and the behaviors of these animals. This will reveal if the rCeA and cCeA are two distinct sub-nuclei or a continuous structure with a subtle gradient.

Thank you for the suggestion. We are inclined to regard the CeA as a continuous structure with a subtle gradient. As our dataset does not include any animals targeting the middle a simple correlation would be unable to reveal the entire relationship between phenotype and fiber placement (it would reveal two clusters best fit by a line, with an empty gap in the middle as in Extended Data Fig. 5d). However, several of our other studies involving anterograde and retrograde tracing involved variable expression distributions across the CeA. These reveal what appear to be a continuous distribution with smoothly varying relationship to various outcomes such as PBN connection strength (Extended Data Fig. 1f) and SI and BNST connectivity (Extended Data Fig. 3c-d). We were interested in what this gradient in connectivity might suggest functionally, so we specifically targeted the extremes of the distribution for manipulation. This allows us to understand the maximum likely differences to be observed biologically. We expect that under normal conditions, activity is balanced across the gradient to push behavior towards these different possibilities, as we describe in the discussion (lines 307-310), and now also explicitly mention as an important area of study in future (lines 339-340).

2, My concern is still the definitions of rCeA and cCeA in this study. While I appreciate that the authors precisely controlled the stereotaxic coordinates, viral injection volumes, and expression levels, it still seems complicated to distinguish such small subregions. Because previous studies, including the authors' own, have demonstrated many cell-type specific marker molecules, such as PKC delta, Somatostatin, and CRF, I recommend verifying the cell types in different coordinates using immunostaining or other methods.

Thank you for the suggestion, we agree that identifying methods for isolating the topography of the CeA outside of targeted injections is crucial for others to leverage these findings in future work. Ours and others' previous work identified a gradient of PKC δ and CRH expression across the CeA (Han et al. 2015, Kim et al. 2017, Sanford et al. 2017). We have added an additional experiment explicitly comparing expression of various cell-type markers in the rostral vs caudal CeA. We find that there are several genes co-expressed with *Calcr1* with significant spatial expression differences, with *Drd2* co-expressed preferentially in the rCeA and *Tacr1/Pkcd* preferentially co-expressed caudally (new Extended Data Fig. 2, lines 111-119). These findings may be used in the future to aid targeting rostral vs. caudal CeA populations with a single viral injection.

3, The response to Reviewer #1 point #14 was rather confusing to this reviewer. They found that the photoinhibition of either rCeA or cCeA *Calcr1* neurons during the foot shock delivery had no effect on fear learning (Extended Data Fig. 9i-j). The authors argue that these results suggest that while rCeA *Calcr1* neuron activity following the US conveys stimulus valence and promotes behavioral responses, the activity is not necessary for downstream associative

processing. However, their previous work (Han et al., 2015) demonstrated that silencing Calcr1 neurons in the CeA significantly attenuated fear learning. Are these potentially opposing results due to methodological differences (TeTx vs. GtACR2) or potential compensatory roles of rCeA and cCeA? Does silencing both rCeA and cCeA during the foot shock attenuate the fear learning?

These apparent differences in findings can be resolved if the methodological differences are considered. TeTx expression in the study by Han et al. differed from our GtACR2 study in multiple ways. First, it silenced all Calcr1 neurons across the entire CeA, second, it silenced Calcr1 neurons both **before, during, and after** memory formation. Hence, this manipulation is unable to resolve whether activity of Calcr1 neurons contributes to US encoding enabling memory formation, or consolidation to reinforce an association, or generation of the learned fear behavior (or all of the above). Our TeTx inhibition study silencing either the rostral or caudal CeA Calcr1+ neurons was most comparable to the study by Han et al.; in this case we replicated their findings (Fig. 5o-p). On the other hand, our GtACR2 study tested the specific question of whether activity in rostral or caudal CeA Calcr1+ neurons **during the foot shock** was specifically important for fear learning. In this case, neural activity was only affected for a total of 32 s, for 4 s each around each of the 8 conditioning foot shocks. In this case, fear learning was unaffected. This suggests that relay of the foot shock through CeA Calcr1+ neurons is not necessary for fear memory formation but does not account for whether their activity is important for consolidation or fear recall.

4, The response to my point #8 was not clear. The authors' response to my comment says, "the bulk of the cells are located quite ventrally, over 1 mm deeper than our fiber tips in this stimulation experiment. Because of this, and the variability in fiber placement that results in locations slightly rostral or caudal from this AP level, we do not think that this is the reason for the difference in behavior in the place-aversion experiment. Additionally, these same animals were used in the conditioned place aversion study where cCeA stimulation led to enhanced place aversion."

The author's group previously demonstrated that Calcr1 neurons are more in rCeA, and stimulating Calcr1 neurons in the CeA induces freezing behaviors and fear learning (Han et al., 2015). Do the authors suggest that the Calcr1 neurons stimulated in the present study are different from those in the previous work by Han et al. because they reside deeper (over 1 mm deeper than our fiber tips)? Or are there distinct sub-regions of CeA along the DV axis as well?

We agree with the primary point that there are sections of the rostral CeA with greater numbers of Calcr1 neurons than some sections of the caudal CeA. Our argument related to the probability that this difference in cell number could be responsible for our observed phenotypes. We argue that, since the cell density is comparable near the fiber tips in both placement conditions (the CeA is similarly wide/tall in the dorsal regions we targeted our fibers towards, but enlarges ventrally in rostral regions), that the delivered light would activate similar numbers of neurons in both scenarios. Regarding the question of sub-regions along the DV axis, our studies sought to disentangle rostral vs. caudal so we carefully maintained DV targeting to minimize variability outside the area of explicit inquiry. Hence, we cannot speak as to whether such a distinction exists. This could be a fruitful area of research for future study.

REVIEWERS' COMMENTS

Reviewer #2 (Remarks to the Author):

The authors appropriately addressed all of my comments. I support the publication.